# THE RELATIVISTIC DISCRIMINATOR: A KEY ELEMENT MISSING FROM STANDARD GAN

**Alexia Jolicoeur-Martineau**
Lady Davis Institute
MILA, Université de Montréal
Montreal, Canada
`alexia.jolicoeur-martineau@mail.mcgill.ca`

## ABSTRACT

In standard generative adversarial network (SGAN), the discriminator $D$ estimates the probability that the input data is real. The generator $G$ is trained to increase the probability that fake data is real. We argue that it should also simultaneously decrease the probability that real data is real because 1) this would account for *a priori* knowledge that half of the data in the mini-batch is fake, 2) this would be observed with divergence minimization, and 3) SGAN would be more similar to integral probability metric (IPM) GANs.

We show that this property can be induced by using a "relativistic discriminator" which estimate the probability that the given real data is more realistic than a randomly sampled fake data. We also present a variant in which the discriminator estimate the probability that the given real data is more realistic than fake data, on average. We generalize both approaches to non-standard GAN loss functions and we refer to them respectively as Relativistic GANs (RGANs) and Relativistic average GANs (RaGANs). We show that IPM-based GANs are a subset of RGANs which use the identity function.

Empirically, we observe that 1) RGANs and RaGANs are significantly more stable and generate higher quality data samples than their non-relativistic counterparts, 2) Standard RaGAN with gradient penalty generate data of better quality than WGAN-GP while only requiring a single discriminator update per generator update (reducing the time taken for reaching the state-of-the-art by 400%), and 3) RaGANs are able to generate plausible high resolutions images (256x256) from a very small sample (N=2011), while GAN and LSGAN cannot; these images are of significantly better quality than the ones generated by WGAN-GP and SGAN with spectral normalization.

The code is freely available on https://github.com/AlexiaJM/RelativisticGAN.

## 1    INTRODUCTION

Generative adversarial networks (GANs) (Hong et al., 2017) form a broad class of generative models in which a game is played between two competing neural networks, the discriminator $D$ and the generator $G$. $D$ is trained to discriminate real from fake data, while $G$ is trained to generate fake data that $D$ will mistakenly recognize as real. In the original GAN by Goodfellow et al. (2014), which we refer to as Standard GAN (SGAN), $D$ is a classifier, thus it is predicting the probability that the input data is real. When $D$ is optimal, the loss function of SGAN is approximately equal to the Jensen–Shannon divergence (JSD) (Goodfellow et al., 2014).

SGAN has two variants for the generator loss functions: saturating and non-saturating. In practice, the former has been found to be very unstable, while the latter has been found to more stable (Goodfellow et al., 2014). Under certain conditions, Arjovsky & Bottou (2017) proved that, if real and fake data are perfectly classified, the saturating loss has zero gradient and the non-saturating loss has non-zero, but volatile gradient. In practice, this means that the discriminator in SGAN often cannot be trained to optimality or with a too high learning rate; otherwise, gradients may vanish and, if so, training will stop. This problem is generally more noticeable in high-dimensional setting (e.g., high resolution

images and discriminator architectures with high expressive power) given that there are enough degrees of freedom available to perfectly classify the training set.

To improve on SGAN, many GAN variants have been suggested using different loss functions and discriminators that are not classifiers (e.g., LSGAN (Mao et al., 2017), WGAN (Arjovsky et al., 2017)). Although these approaches have partially succeeded in improving stability and data quality, the large-scale study by Lucic et al. (2017) suggests that these approaches do not consistently improve on SGAN. Additionally, some of the most successful approaches, such as WGAN-GP (Gulrajani et al., 2017), are much more computationally demanding than SGAN.

Many of the recent successful GANs variants have been based on Integral probability metrics (IPMs) (Müller, 1997) (e.g., WGAN , WGAN-GP, Fisher GAN (Mroueh & Sercu, 2017), Sobolev GAN (Mroueh et al., 2017)). In IPM-based GANs, the discriminator is real-valued and constrained to a specific class of function which regularize the discriminator. See Mroueh et al. (2017) for a review of the different IPMs.

These IPM constraints have been shown to be beneficial even in non-IPM based GANs. Spectral normalization (Miyato et al., 2018) improves the stability of various GANs and it consists in making the discriminator Lipschitz-1, which is the constraint of WGAN. Similarly, the gradient penalty of WGAN-GP also provides improve the stability of SGAN (Fedus et al., 2017). Although this shows that certain IPM constraints improve the stability of GANs, it does not explain why IPM-based GANs generally provide increased stability over other metrics/divergences in GANs (e.g., JSD for SGAN, $f$-divergences for $f$-GANs (Nowozin et al., 2016)).

Note that although powerful, IPM-based GANs tend to more computationally demanding than other GANs. Certain IPM-based GANs use a gradient penalty (e.g. WGAN-GP, Sobolev GAN) which is very computationally costly and most IPM-based GANs need more than one discriminator update per generator update (WGAN-GP requires at least 5 (Gulrajani et al., 2017)). Assuming equal training time for $D$ and $G$, every additional discriminator update increase training time by a significant 50%.

In this paper, we argue that non-IPM-based GANs are missing a key ingredient, a relativistic discriminator, which IPM-based GANs already possess. We show that a relativistic discriminator is necessary to make GANs analogous to divergence minimization and produce sensible predictions based on the *a priori* knowledge that half of the samples in the mini-batch are fake. We provide empirical evidence showing that GANs with a relativistic discriminator are more stable and produce data of higher quality.

## 2 BACKGROUND

### 2.1 GENERATIVE ADVERSARIAL NETWORKS

GANs can be defined very generally in terms of the discriminator in the following way:

$$L_D = \mathbb{E}_{x_r \sim \mathbb{P}} \left[ \tilde{f}_1(D(x_r)) \right] + \mathbb{E}_{z \sim \mathbb{P}_z} \left[ \tilde{f}_2(D(G(z))) \right], \tag{1}$$

and

$$L_G = \mathbb{E}_{x_r \sim \mathbb{P}} \left[ \tilde{g}_1(D(x_r)) \right] + \mathbb{E}_{z \sim \mathbb{P}_z} \left[ \tilde{g}_2(D(G(z))) \right], \tag{2}$$

where $\tilde{f}_1, \tilde{f}_2, \tilde{g}_1, \tilde{g}_2$ are scalar-to-scalar functions, $\mathbb{P}$ is the distribution of real data, $\mathbb{P}_z$ is generally a multivariate normal distribution centered at 0 with variance 1, $D(x)$ is the discriminator evaluated at $x$, $G(z)$ is the generator evaluated at z ($\mathbb{Q}$ is the distribution of fake data, thus of $G(z)$). Note that, through the paper, we refer to real data as $x_r$ and fake data as $x_f$. Without loss of generality, we assume that both $L_D$ and $L_G$ are loss functions to be minimized.

Most GANs can be separated into two classes: non-saturating and saturating loss functions. GANs with the saturating loss are such that $\tilde{g}_1 = -\tilde{f}_1$ and $\tilde{g}_2 = -\tilde{f}_2$, while GANs with the non-saturating loss are such that $\tilde{g}_1 = \tilde{f}_2$ and $\tilde{g}_2 = \tilde{f}_1$. Saturating GANs are intuitive as they can be interpreted as alternating between maximizing and minimizing the same loss function. After training $D$ to optimality, the loss function is generally an approximation of a divergence (e.g., Jensen–Shannon divergence (JSD) for SGAN (Goodfellow et al., 2014), $f$-divergences for F-GANs (Nowozin et al., 2016), and Wassertein distance for WGAN (Arjovsky et al., 2017)). Thus, training $G$ to minimize $L_G$ can be roughly interpreted as minimizing the approximated divergence (although this is not technically true; see

Jolicoeur-Martineau (2018)). On the other hand, non-saturating GANs can be thought as optimizing the same loss function, but swapping real data with fake data (and vice-versa). In this article, unless otherwise specified, we assume a non-saturating loss for all GANs.

SGAN assumes a cross-entropy loss, i.e., $\tilde{f}_1(D(x)) = -\log(D(x))$ and $\tilde{f}_2(D(x)) = -\log(1 - D(x))$, where $D(x) = \text{sigmoid}(C(x))$, and $C(x)$ is the non-transformed discriminator output (which we call the *critic* as per Arjovsky et al. (2017)). In most GANs, $C(x)$ can be interpreted as *how realistic the input data is*; a negative number means that the input data looks fake (e.g., in SGAN, $D(x) = \text{sigmoid}(-5) = 0$), while a positive number means that the input data looks real (e.g., in SGAN, $D(x) = \text{sigmoid}(5) = 1$).

## 2.2 INTEGRAL PROBABILITY METRICS

IPMs are statistical divergences represented mathematically as:

$$IPM_F(\mathbb{P}||\mathbb{Q}) = \sup_{C \in \mathcal{F}} \mathbb{E}_{x \sim \mathbb{P}}[C(x)] - \mathbb{E}_{x \sim \mathbb{Q}}[C(x)],$$

where $\mathcal{F}$ is a class of real-valued functions.

IPM-based GANs can be defined using equation 1 and 2 where $\tilde{f}_1(D(x)) = \tilde{g}_2(D(x)) = -D(x)$ and $\tilde{f}_2(D(x)) = \tilde{g}_1(D(x)) = D(x)$, where $D(x) = C(x)$ (i.e., no transformation is applied). It can be observed that both discriminator and generator loss functions are unbounded and would diverge to $-\infty$ if optimized directly. However, IPMs assume that the discriminator is of a certain class of function that does not grow too quickly which prevent the loss functions from diverging. Each IPM applies a different constraint to the discriminator (e.g., WGAN assumes a Lipschitz $D$, WGAN-GP assumes that $D$ has a gradient norm equal to 1 around real and fake data).

## 3 MISSING PROPERTY OF SGAN

We argue that the key missing property of SGAN is that the probability of real data being real ($D(x_r)$) should decrease as the probability of fake data being real ($D(x_f)$) increase. We provide three arguments suggesting that SGAN should have this property.

### 3.1 PRIOR KNOWLEDGE ARGUMENT

Assuming a rational human was shown half real data and half fake data. If they perceived all samples shown as equally real ($C(x_f) \approx C(x_r)$ for most $x_r$ and $x_f$), they would assume that each sample has probability .50 of being real. However, this is not the case for the discriminator in SGAN. If all samples looked real ($C(x_f) \approx C(x_r) \geq 3$), $D$ would assume incorrectly that they are indeed all real ($D(x) \approx 1$ for all $x$). Of course, once trained with the labels, $D$ would decrease $D(x_f)$ and thus would obtain more reasonable estimates. However, if $D(x_r)$ decreased as $D(x_f)$ increased, we would have had that $D(x) \approx .50$ for all $x$ before even retraining $D$. A rational human would not require retraining. IPM-based GANs implicitly account for the fact that some of the samples must be fake because they compare how realistic real data is compared to fake data. This is the behavior that we would want.

### 3.2 DIVERGENCE MINIMIZATION ARGUMENT

In SGAN, when optimized, we have that the discriminator loss function is equal to the Jensen–Shannon divergence (JSD) (Goodfellow et al., 2014). The JSD is minimized ($JSD(\mathbb{P}||\mathbb{Q}) = 0$) when $D(x_r) = D(x_f) = \frac{1}{2}$ for all $x_r \in \mathbb{P}$ and $x_f \in \mathbb{Q}$ and maximized ($JSD(\mathbb{P}||\mathbb{Q}) = \log(2)$) when $D(x_r) = 1$, $D(x_f) = 0$ for all $x_r \in \mathbb{P}$ and $x_f \in \mathbb{Q}$. Thus, if we were directly minimizing the divergence from maximum to minimum, we would expect $D(x_r)$ to smoothly decrease from 1 to .50 for most $x_r$ and $D(x_f)$ to smoothly increase from 0 to .50 for most $x_f$ (Figure 1a). However, when minimizing the saturating loss in SGAN, we are only increasing $D(x_f)$, we are not decreasing $D(x_r)$ (Figure 1b). Furthermore, we are bringing $D(x_f)$ closer to 1 rather than .50.

This means that SGAN dynamics are very different from the minimization of the JSD. To bring SGAN closer to divergence minimization, training the generator should not only increase $D(x_f)$ but

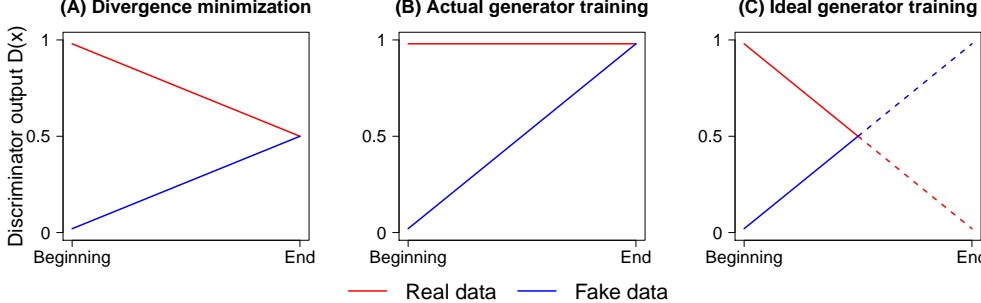

Figure 1: Expected discriminator output of the real and fake data for the a) direct minimization of the Jensen–Shannon divergence, b) actual training of the generator to minimize its loss function, and c) ideal training of the generator to minimize its loss function (lines are dotted when they cross beyond the equilibrium to signify that this may or may not be necessary).

also decrease $D(x_r)$ (Figure 1c). Note that although specific to the JSD, similar dynamics are true for other divergences; when the divergence is maximal, $D(x_r)$ and $D(x_f)$ are very far from one another, but they converge to the same value as the divergence approach zero. Thus, this argument applies to other divergences.

## 3.3 GRADIENT ARGUMENT

We compare the gradients of standard GAN and IPM-based GANs for further insight. It can be shown that the gradients of the discriminator and generator in non-saturating SGAN are respectively:

$$\nabla_w L_D^{GAN} = -\mathbb{E}_{x_r \sim \mathbb{P}} \left[ (1 - D(x_r)) \nabla_w C(x_r) \right] + \mathbb{E}_{x_f \sim \mathbb{Q}_\theta} \left[ D(x_f) \nabla_w C(x_f) \right], \tag{3}$$

$$\nabla_\theta L_G^{GAN} = -\mathbb{E}_{z \sim \mathbb{P}_z} \left[ (1 - D(G(z))) \nabla_x C(G(z)) J_\theta G(z) \right], \tag{4}$$

where $J$ is the Jacobian.

It can be shown that the gradients of the discriminator and generator in IPM-based GANs are respectively:

$$\nabla_w L_D^{IPM} = -\mathbb{E}_{x_r \sim \mathbb{P}} [\nabla_w C(x_r)] + \mathbb{E}_{x_f \sim \mathbb{Q}_\theta} [\nabla_w C(x_f)], \tag{5}$$

$$\nabla_\theta L_G^{IPM} = -\mathbb{E}_{z \sim \mathbb{P}_z} [\nabla_x C(G(z)) J_\theta G(z)], \tag{6}$$

where $C(x) \in \mathcal{F}$ (the class of functions assigned by the IPM).

From these equations, it can be observed that SGAN leads to the same dynamics as IPM-based GANs when we have that:

1. $D(x_r) = 0$, $D(x_f) = 1$ in the discriminator step of SGAN
2. $D(x_f) = 0$ in the generator step of SGAN.
3. $C(x) \in \mathcal{F}$

Assuming that the discriminator and generator are trained to optimality in each step (which we sometimes do for $D$, but never for $G$) and that it is possible to perfectly distinguish real from the fake data (strong assumption, but generally true early in training); we would have that $D(x_r) = 1$, $D(x_f) = 0$ in the generator step and that $D(x_r) = 1$, $D(x_f) = 1$ in the discriminator step for most $x_r$ and $x_f$ (Figure 1b). Thus, the only missing assumption would be that $D(x_r) = 0$ in the discriminator step.

Although the above scenario is not realistic (because we never train $G$ to optimality), if all the assumptions were respected and the generator could indirectly influence $D(x_r)$, we would have that $D(x_r) = 0$, $D(x_f) = 1$. Thus, SGAN would have the same gradients as IPM-based GANs. We conjecture that making SGAN more similar to IPM-based GANs could potentially improve its stability (our results will show that it does in fact improves stability).

## 4 METHOD

### 4.1 RELATIVISTIC STANDARD GAN

In standard GAN, the discriminator can be defined, in term of the non-transformed layer $C(x)$, as $D(x) = \text{sigmoid}(C(x))$. A simple way to make discriminator relativistic (i.e., having the output of $D$ depends on both real and fake data) is to sample from real/fake data pairs $\tilde{x} = (x_r, x_f)$ and define it as $D(\tilde{x}) = \text{sigmoid}(C(x_r) - C(x_f))$.

We can interpret this modification in the following way: ***the discriminator estimates the probability that the given real data is more realistic than a randomly sampled fake data***. Similarly, we can define $D_{rev}(\tilde{x}) = \text{sigmoid}(C(x_f) - C(x_r))$ as the probability that the given fake data is more realistic than a randomly sampled real data. An interesting property of this discriminator is that we do not need to include $D_{rev}$ in the loss function through $\log(1 - D_{rev}(\tilde{x}))$ because we have that $1 - D_{rev}(\tilde{x}) = 1 - \text{sigmoid}(C(x_f) - C(x_r)) = \text{sigmoid}(C(x_r) - C(x_f)) = D(\tilde{x})$; thus, $\log(D(\tilde{x})) = \log(1 - D_{rev}(\tilde{x}))$.

The discriminator and generator (non-saturating) loss functions of the Relativistic Standard GAN (RSGAN) can be written as:

$$L_D^{RSGAN} = -\mathbb{E}_{(x_r, x_f) \sim (\mathbb{P}, \mathbb{Q})} \left[ \log(\text{sigmoid}(C(x_r) - C(x_f))) \right]. \tag{7}$$

$$L_G^{RSGAN} = -\mathbb{E}_{(x_r, x_f) \sim (\mathbb{P}, \mathbb{Q})} \left[ \log(\text{sigmoid}(C(x_f) - C(x_r))) \right]. \tag{8}$$

### 4.2 RELATIVISTIC GANS

More generally, we consider any discriminator defined as $a(C(x_r) - C(x_f))$, where $a$ is the activation function, to be relativistic. This means that almost any GAN can have a relativistic discriminator. This forms a new class of models which we call Relativistic GANs (RGANs).

Most GANs can be parametrized very generally in terms of the critic:

$$L_D^{GAN} = \mathbb{E}_{x_r \sim \mathbb{P}} \left[ f_1(C(x_r)) \right] + \mathbb{E}_{x_f \sim \mathbb{Q}} \left[ f_2(C(x_f)) \right] \tag{9}$$

and

$$L_G^{GAN} = \mathbb{E}_{x_r \sim \mathbb{P}} \left[ g_1(C(x_r)) \right] + \mathbb{E}_{x_f \sim \mathbb{Q}} \left[ g_2(C(x_f)) \right], \tag{10}$$

where $f_1$, $f_2$, $g_1$, $g_2$ are scalar-to-scalar functions. If we use a relativistic discriminator, these GANs now have the following form:

$$L_D^{RGAN} = \mathbb{E}_{(x_r, x_f) \sim (\mathbb{P}, \mathbb{Q})} \left[ f_1(C(x_r) - C(x_f)) \right] + \mathbb{E}_{(x_r, x_f) \sim (\mathbb{P}, \mathbb{Q})} \left[ f_2(C(x_f) - C(x_r)) \right] \tag{11}$$

and

$$L_G^{RGAN} = \mathbb{E}_{(x_r, x_f) \sim (\mathbb{P}, \mathbb{Q})} \left[ g_1(C(x_r) - C(x_f)) \right] + \mathbb{E}_{(x_r, x_f) \sim (\mathbb{P}, \mathbb{Q})} \left[ g_2(C(x_f) - C(x_r)) \right]. \tag{12}$$

If one use the identity function (i.e., $f_1(y) = g_2(y) = -y$, $f_2(y) = g_1(y) = y$), this results in a degenerate case since there is no supremum/maximum. However, if one adds a constraint so that $C(x_r) - C(x_f)$ is bounded, then there is a supremum and one arrives at IPM-based GANs. Thus, although different, IPM-based GANs share a very similar loss function focused on the difference in critics.

Importantly, $g_1$ is normally ignored in GANs because its gradient is zero since the generator does not influence it. However, in RGANs, $g_1$ is influenced by fake data, thus by the generator.

### 4.3 RELATIVISTIC AVERAGE GANS

The discriminator has a very different interpretation in SGAN compared to RSGAN. In SGAN, $D(x)$ estimates the probability that $x$ is real, while in RGANs, $D(x_r, x_f)$ estimates the probability that $x_r$ is more realistic than $x_f$. As a middle ground, we developed an alternative to the Relativistic Discriminator, which retains approximately the same interpretation as the discriminator in SGAN while still being relativistic.

We propose the Relativistic average Discriminator (RaD) which compares the critic of the input data to the average critic of samples of the opposite type. The discriminator loss function for this approach can be formulated as:

$$L_D^{RaSGAN} = -\mathbb{E}_{x_r \sim \mathbb{P}} \left[ \log \left( \bar{D}(x_r) \right) \right) \right] - \mathbb{E}_{x_f \sim \mathbb{Q}} \left[ \log \left( 1 - \bar{D}(x_f) \right) \right], \tag{13}$$

where

$$\bar{D}(x) = \begin{cases} \text{sigmoid}(C(x) - \mathbb{E}_{x_f \sim \mathbb{Q}} C(x_f)) & \text{if } x \text{ is real} \\ \text{sigmoid}(C(x) - \mathbb{E}_{x_r \sim \mathbb{P}} C(x_r)) & \text{if } x \text{ is fake.} \end{cases} \tag{14}$$

RaD has a more similar interpretation to the standard discriminator than the relativistic discriminator. With RaD, ***the discriminator estimates the probability that the given real data is more realistic than fake data, on average***.

As before, we can generalize this approach to work with any GAN loss function using the following formulation:

$$L_D^{RaGAN} = \mathbb{E}_{x_r \sim \mathbb{P}} \left[ f_1 \left( C(x_r) - \mathbb{E}_{x_f \sim \mathbb{Q}} C(x_f) \right) \right] + \mathbb{E}_{x_f \sim \mathbb{Q}} \left[ f_2 \left( C(x_f) - \mathbb{E}_{x_r \sim \mathbb{P}} C(x_r) \right) \right]. \tag{15}$$

$$L_G^{RaGAN} = \mathbb{E}_{x_r \sim \mathbb{P}} \left[ g_1 \left( C(x_r) - \mathbb{E}_{x_f \sim \mathbb{Q}} C(x_f) \right) \right] + \mathbb{E}_{x_f \sim \mathbb{Q}} \left[ g_2 \left( C(x_f) - \mathbb{E}_{x_r \sim \mathbb{P}} C(x_r) \right) \right]. \tag{16}$$

We call this general approach Relativistic average GAN (RaGAN).

## 5 EXPERIMENTS

Experiments were conducted on the CIFAR-10 dataset (Krizhevsky, 2009) and the CAT dataset (Zhang et al., 2008). Code was written in Pytorch (Paszke et al., 2017) and models were trained using the Adam optimizer (Kingma & Ba, 2014) for 100K generator iterations with seed 1 (which shows that we did not fish for the best seed, instead, we selected the seed *a priori*). We report the Fréchet Inception Distance (FID) (Heusel et al., 2017), a measure that is generally better correlated with data quality than the Inception Distance (Salimans et al., 2016) (Borji, 2018); lower FID means that the generated images are of better quality.

For the models architectures, we used the standard CNN described by Miyato et al. (2018) on CIFAR-10 and a relatively standard DCGAN architecture (Radford et al., 2015) on CAT (see Appendix). We also provide the source code required to replicate all analyses presented in this paper (See our repository: [Anonymous until peer review is finished]).

### 5.1 CIFAR-10

In these analyses, we compared standard GAN (SGAN), least-squares GAN (LSGAN), Wassertein GAN improved (WGAN-GP), Hinge-loss GAN (HingeGAN) (Miyato et al., 2018), Relativistic SGAN (RSGAN), Relativistic average SGAN (RaSGAN), Relativistic average LSGAN (RaLSGAN), and Relativistic average HingeGAN (RaHingeGAN) using the standard CNN architecture on stable setups (See Appendix for details on the loss functions used). Additionally, we tested RSGAN and RaSGAN with the same gradient-penalty as WGAN-GP (named RSGAN-GP and RaSGAN-GP respectively).

We used the following two known stable setups: (DCGAN setup) $lr = .0002$, $n_D = 1$, $\beta_1 = .50$ and $\beta_2 = .999$ (Radford et al., 2015), and (WGAN-GP setup) $lr = .0001$, $n_D = 5$, $\beta_1 = .50$ and $\beta_2 = .9$ (Gulrajani et al., 2017), where $lr$ is the learning rate, $n_D$ is the number of discriminator updates per generator update, and $\beta_1$, $\beta_2$ are the ADAM momentum parameters. For optimal stability, we used batch norm (Ioffe & Szegedy, 2015) in $G$ and spectral norm (Miyato et al., 2018) in $D$.

Results are presented in Table 1. We observe that RSGAN and RaSGAN generally performed better than SGAN. Similarly, RaHingeGAN performed better than HingeGAN. RaLSGAN performed on par with LSGAN, albeit sightly worse. WGAN-GP performed poorly in the DCGAN setup, but very well in the WGAN-GP setup. RasGAN-GP performed poorly; however, RSGAN-GP performed better than all other loss functions using only one discriminator update per generator update. Importantly, the resulting FID of 25.60 is on par with the lowest FID obtained for this architecture using spectral normalization, as reported by Miyato et al. (2018) (25.5). Overall, these results show that using a relativistic discriminator generally improve data generation quality and that RSGAN works very well in conjunction with gradient penalty to obtain state-of-the-art results.

Table 1: Fréchet Inception Distance (FID) at exactly 100k generator iterations on the CIFAR-10 dataset using stable setups with different GAN loss functions. We used spectral norm in $D$ and batch norm in $G$. All models were trained using the same *a priori* selected seed (seed=1).

| Loss | $lr = .0002$ $\beta = (.50, .999)$ $n_D = 1$ | $lr = .0001$ $\beta = (.50, .9)$ $n_D = 5$ |
|---|---|---|
| SGAN | 40.64 | 41.32 |
| RSGAN | 36.61 | 55.29 |
| RaSGAN | 31.98 | 37.92 |
| LSGAN | 29.53 | 187.01 |
| RaLSGAN | 30.92 | 219.39 |
| HingeGAN | 49.53 | 80.85 |
| RaHingeGAN | 39.12 | 37.72 |
| WGAN-GP | 83.89 | **27.81** |
| RSGAN-GP | **25.60** | 28.13 |
| RaSGAN-GP | 331.86 | |

## 5.2 CAT

CAT is a dataset containing around 10k pictures of cats with annotations. We cropped the pictures to the faces of the cats using those annotations. After removing outliers (hidden faces, blurriness, etc.), the CAT dataset contained 9304 images $\geq$ 64x64, 6645 images $\geq$ 128x128, and 2011 images $\geq$ 256x256. The CAT dataset is particularly challenging due to its small sample size and high-resolution images; this makes it perfect for testing the stability of different GAN loss functions.

We trained different GAN loss functions on 64x64, 128x128, 256x256 images. For 256x256 images, we compared RaGANs to known stable approaches: SpectralSGAN (SGAN with spectral normalization in $D$) and WGAN-GP. Although some approaches were able to train on 256x256 images, they did so with significant mode collapse. To alleviate this problem, for 256x256 images, we packed the discriminator (Lin et al., 2017) (i.e., $D$ took a concatenated pair of images instead of a single image). We looked at the minimum, maximum, mean and standard deviation (SD) of the FID at 20k, 30k, ..., 100k generator iterations; results are presented in Table 2.

Overall, we observe lower minimum FID, maximum FID, mean and standard deviation (sd) for RGANs and RaGANs than their non-relativistic counterparts (SGAN, LSGAN, RaLSGAN).

In 64x64 resolution, both SGAN and LSGAN generated images with low FID, but they did so in a very unstable matter. For example, SGAN went from a FID of 17.50 at 30k iterations, to 310.56 at 40k iterations, and back to 27.72 at 50k iterations. Similarly, LSGAN went from a FID of 20.27 at 20k iterations, to 224.97 at 30k iterations, and back to 51.98 at 40k iterations. On the other hand, RaGANs were much more stable (lower max and SD) while also resulting in lower minimum FID. Using gradient-penalty did not improve data quality; however, it reduced the SD lower than without gradient penalty, thus increasing stability further.

SGAN was unable to converge on 128x128 or bigger images and LSGAN was unable to converge on 256x256 images. Meanwhile, RaGANs were able to generate plausible images with low FID in all resolutions. Although SpectralSGAN and WGAN-GP were able to generate 256x256 images of cats, the samples they generated were of poor quality (high FID). Thus, in this very difficult setting, relativism provided a greater improvement in quality than gradient penalty or spectral normalization.

## 6 CONCLUSION AND FUTURE WORK

In this paper, we proposed the relativistic discriminator as a way to fix and improve on standard GAN. We further generalized this approach to any GAN loss and introduced a generally more stable variant called RaD. Our results suggest that relativism significantly improve data quality and stability of

Table 2: Minimum (min), maximum (max), mean, and standard deviation (SD) of the Fréchet Inception Distance (FID) calculated at 20k, 30k ..., 100k generator iterations on the CAT dataset with different GAN loss functions. The hyper-parameters used were $lr = .0002$, $\beta = (.50, .999)$, $n_D = 1$, and batch norm (BN) in $D$ and $G$. All models were trained using the same *a priori* selected seed (seed=1). Note: A missing number imply that the model did not converge and became stuck in the first few iterations.

| Loss | Min | Max | Mean | SD |
|---|---|---|---|---|
| 64x64 images (N=9304) | | | | |
| SGAN | 16.56 | 310.56 | 52.54 | 96.81 |
| RSGAN | 19.03 | 42.05 | 32.16 | 7.01 |
| RaSGAN | 15.38 | 33.11 | 20.53 | 5.68 |
| LSGAN | 20.27 | 224.97 | 73.62 | 61.02 |
| RaLSGAN | **11.97** | **19.29** | **15.61** | 2.55 |
| HingeGAN | 17.60 | 50.94 | 32.23 | 14.44 |
| RaHingeGAN | 14.62 | 27.31 | 20.29 | 3.96 |
| RSGAN-GP | 16.41 | 22.34 | 18.20 | 1.82 |
| RaSGAN-GP | 17.32 | 22 | 19.58 | **1.81** |
| 128x128 images (N=6645) | | | | |
| SGAN | - | - | - | - |
| RaSGAN | 21.05 | **39.65** | 28.53 | **6.52** |
| LSGAN | 19.03 | 51.36 | 30.28 | 10.16 |
| RaLSGAN | **15.85** | 40.26 | **22.36** | 7.53 |
| 256x256 images (N=2011) | | | | |
| SGAN[1] | - | - | - | - |
| RaSGAN | **32.11** | 102.76 | **56.64** | 21.03 |
| SpectralSGAN | 54.08 | **90.43** | 64.92 | **12.00** |
| LSGAN[1] | - | - | - | - |
| RaLSGAN | 35.21 | 299.52 | 70.44 | 86.01 |
| WGAN-GP | 155.46 | 437.48 | 341.91 | 101.11 |

GANs at no computational cost. Furthermore, using a relativistic discriminator with other tools of the trade (spectral norm, gradient penalty, etc.) may lead to better state-of-the-art.

Future research is needed to fully understand the mathematical implications of adding relativism to GANs. Furthermore, our experiments were limited to certain loss functions using only one seed, due to computational constraints. More experiments are required to determine which relativistic GAN loss function is best over a wide-range of datasets and hyper-parameters. We greatly encourage researchers and machine learning enthusiasts with greater computing power to experiment further with our approach.

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

## APPENDICES

### A  INTUITIVE AND MEMEFUL VISUAL REPRESENTATION OF RAGANS

Table 3: An illustrative example of the discriminator's output in standard GAN as traditionally defined $(P(x_r \text{ is real}) = \text{sigmoid}(C(x_r)))$ versus the Relativistic average Discriminator (RaD) $(P(x_r \text{ is real}|\overline{C(x_f)}) = \text{sigmoid}(C(x_r) - \overline{C(x_f)}))$. Breads represent real images, while dogs represent fake images.

| Scenario | Absolute probability (Standard GAN) | Relative probability (Relativistic average Standard GAN) |
|---|---|---|
| Real image looks real **and** fake images look fake |  $C(x_r) = 8$ $P(x_r \text{ is bread}) = 1$ |  $\overline{C(x_f)} = -5$ $P(x_r \text{ is bread}|\overline{C(x_f)}) = 1$ |
| Real image looks real **but** fake images look similarly real on average |  $C(x_r) = 8$ $P(x_r \text{ is bread}) = 1$ |  $\overline{C(x_f)} = 7$ $P(x_r \text{ is bread}|\overline{C(x_f)}) = .73$ |
| Real image looks fake **but** fake images look more fake on average |  $C(x_r) = -3$ $P(x_r \text{ is bread}) = .05$ |  $\overline{C(x_f)} = -5$ $P(x_r \text{ is bread}|\overline{C(x_f)}) = .88$ |

### B  INTUITION BEHIND RAGANS

Although the relative discriminator provide the missing property that we want in GANs (i.e., $G$ influencing $D(x_r)$), its interpretation is different from the standard discriminator. Rather than measuring "the probability that the input data is real", it is now measuring "the probability that the

input data is more realistic than a randomly sampled data of the opposing type (fake if the input is real or real if the input is fake)". To make the relativistic discriminator act more globally, as in its original definition, our initial idea was the following: average the relativistic discriminator over random samples of data of the opposing type. This can be conceptualized in the following way:

$$
\begin{aligned}
P(x_r \text{ is real}) &:= \mathbb{E}_{x_f \sim \mathbb{Q}}[P(x_r \text{ is more real than } x_f)] \\
&= \mathbb{E}_{x_f \sim \mathbb{Q}}[\text{sigmoid}(C(x_r) - C(x_f))] \\
&= \mathbb{E}_{x_f \sim \mathbb{Q}}[D(x_r, x_f)],
\end{aligned}
$$

$$
\begin{aligned}
P(x_f \text{ is real}) &:= \mathbb{E}_{x_r \sim \mathbb{P}}[P(x_f \text{ is more real than } x_r)] \\
&= \mathbb{E}_{x_r \sim \mathbb{P}}[\text{sigmoid}(C(x_f) - C(x_r))] \\
&= \mathbb{E}_{x_r \sim \mathbb{P}}[D(x_f, x_r)],
\end{aligned}
$$

where $D(x_r, x_f) = \text{sigmoid}(C(x_r) - C(x_f))$.

Then, the following loss function for $D$ could be applied:

$$
L_D = -\mathbb{E}_{x_r \sim \mathbb{P}}\left[\log\left(\mathbb{E}_{x_f \sim \mathbb{Q}}[D(x_r, x_f)])\right)\right] - \mathbb{E}_{x_f \sim \mathbb{Q}}\left[\log\left(1 - \mathbb{E}_{x_r \sim \mathbb{P}}[D(x_f, x_r)]\right)\right]. \tag{17}
$$

The main problem with this idea is that it would require looking at all possible combinations of real and fake data in the mini-batch. This would transform the problem from $\mathcal{O}(m)$ to $\mathcal{O}(m^2)$ complexity, where $m$ is the batch size. This is problematic; therefore, we do not use this approach.

Instead, we propose to use the Relativistic average Discriminator (RaD) which compares the critic of the input data to the average critic of samples of the opposite type (See section 4.3). This approach has $\mathcal{O}(m)$ complexity.

## C  GRADIENTS

### C.1  SGAN

$$
\begin{aligned}
\nabla_w L_D^{GAN} &= -\nabla_w \mathbb{E}_{x_r \sim \mathbb{P}}\left[\log D(x_r)\right] - \nabla_w \mathbb{E}_{x_f \sim \mathbb{Q}_\theta}\left[\log(1 - D(x_f))\right] \\
&= -\nabla_w \mathbb{E}_{x_r \sim \mathbb{P}}\left[\log\left(\frac{e^{C(x_r)}}{e^{C(x_r)} + 1}\right)\right] - \nabla_w \mathbb{E}_{x_f \sim \mathbb{Q}_\theta}\left[\log\left(1 - \frac{e^{C(x_f)}}{e^{C(x_f)} + 1}\right)\right] \\
&= -\nabla_w \mathbb{E}_{x_r \sim \mathbb{P}}\left[C(x_r) - \log\left(e^{C(x_r)} + 1\right)\right] - \nabla_w \mathbb{E}_{x_f \sim \mathbb{Q}_\theta}\left[\log(1) - \log\left(e^{C(x_f)} + 1\right)\right] \\
&= -\mathbb{E}_{x_r \sim \mathbb{P}}\left[\nabla_w C(x_r)\right] + \mathbb{E}_{x_r \sim \mathbb{P}}\left[\frac{e^{C(x_r)}}{e^{C(x_r)} + 1}\nabla_w C(x_r)\right] + \mathbb{E}_{x_f \sim \mathbb{Q}_\theta}\left[\frac{e^{C(x_f)}}{e^{C(x_f)} + 1}\nabla_w C(x_f)\right] \\
&= -\mathbb{E}_{x_r \sim \mathbb{P}}\left[\nabla_w C(x_r)\right] + \mathbb{E}_{x_r \sim \mathbb{P}}\left[D(x_r)\nabla_w C(x_r)\right] + \mathbb{E}_{x_f \sim \mathbb{Q}_\theta}\left[D(x_f)\nabla_w C(x_f)\right] \\
&= -\mathbb{E}_{x_r \sim \mathbb{P}}\left[(1 - D(x_r))\nabla_w C(x_r)\right] + \mathbb{E}_{x_f \sim \mathbb{Q}_\theta}\left[D(x_f)\nabla_w C(x_f)\right]
\end{aligned}
$$

$$
\begin{aligned}
\nabla_\theta L_G^{GAN} &= -\nabla_\theta \mathbb{E}_{z \sim \mathbb{P}_z}\left[\log D(G(z))\right] \\
&= -\nabla_\theta \mathbb{E}_{z \sim \mathbb{P}_z}\left[\log\left(\frac{e^{C(G(z))}}{e^{C(G(z))} + 1}\right)\right] \\
&= -\nabla_\theta \mathbb{E}_{z \sim \mathbb{P}_z}\left[C(G(z)) - \log\left(e^{C(G(z))} + 1\right)\right] \\
&= -\mathbb{E}_{z \sim \mathbb{P}_z}\left[\nabla_x C(G(z))J_\theta G(z) - \left(\frac{e^{C(G(z))}}{e^{C(G(z))} + 1}\right)\nabla_x C(G(z))J_\theta G(z)\right] \\
&= -\mathbb{E}_{z \sim \mathbb{P}_z}\left[(1 - D(G(z)))\nabla_x C(G(z))J_\theta G(z)\right]
\end{aligned}
$$

### C.2  IPM-BASED GANS

$$
\begin{aligned}
\nabla_w L_D^{IPM} &= -\nabla_w \mathbb{E}_{x_r \sim \mathbb{P}}[C(x_r)] + \nabla_w \mathbb{E}_{x_f \sim \mathbb{Q}_\theta}[C(x_f)] \\
&= -\mathbb{E}_{x_r \sim \mathbb{P}}[\nabla_w C(x_r)] + \mathbb{E}_{x_f \sim \mathbb{Q}_\theta}[\nabla_w C(x_f)]
\end{aligned}
$$

$$\nabla_\theta L_G^{IPM} = -\nabla_\theta \mathbb{E}_{z\sim\mathbb{P}_z}[C(G(z))]$$
$$= -\mathbb{E}_{z\sim\mathbb{P}_z}[\nabla_x C(G(z)) J_\theta G(z)]$$

## D  SIMPLIFIED FORM OF RELATIVISTIC SATURATING AND NON-SATURATING GANS

The formulation of RGANs can be simplified when we have the following two properties: (1) $f_2(-y) = f_1(y)$ and (2) the generator assumes a non-saturating loss ($g_1(y) = f_2(y)$ and $g_2(y) = f_1(y)$). These two properties are observed in standard GAN, LSGAN using symmetric labels (e.g., -1 and 1), IPM-based GANs, etc. With these two properties, RGANs with non-saturating loss can be formulated simply as:

$$L_D^{RGAN*} = \mathbb{E}_{(x_r,x_f)\sim(\mathbb{P},\mathbb{Q})}\left[f_1(C(x_r) - C(x_f))\right] \tag{18}$$

and

$$L_G^{RGAN*} = \mathbb{E}_{(x_r,x_f)\sim(\mathbb{P},\mathbb{Q})}\left[f_1(C(x_f) - C(x_r))\right]. \tag{19}$$

Assuming $f_2(-y) = f_1(y)$, we have that

$$L_D^{RGAN} = \mathbb{E}_{(x_r,x_f)\sim(\mathbb{P},\mathbb{Q})}\left[f_1(C(x_r) - C(x_f))\right] + \mathbb{E}_{(x_r,x_f)\sim(\mathbb{P},\mathbb{Q})}\left[f_2(C(x_f) - C(x_r))\right]$$
$$= \mathbb{E}_{(x_r,x_f)\sim(\mathbb{P},\mathbb{Q})}\left[f_1(C(x_r) - C(x_f))\right] + \mathbb{E}_{(x_r,x_f)\sim(\mathbb{P},\mathbb{Q})}\left[f_1(C(x_r) - C(x_f))\right]$$
$$= 2\mathbb{E}_{(x_r,x_f)\sim(\mathbb{P},\mathbb{Q})}\left[f_1(C(x_r) - C(x_f))\right].$$

If $g_1(y) = -f_1(y)$ and $g_2(y) = -f_2(y)$ (saturating GAN), we have that

$$L_G^{RGAN-S} = \mathbb{E}_{(x_r,x_f)\sim(\mathbb{P},\mathbb{Q})}\left[g_1(C(x_r) - C(x_f))\right] + \mathbb{E}_{(x_r,x_f)\sim(\mathbb{P},\mathbb{Q})}\left[g_2(C(x_f) - C(x_r))\right]$$
$$= -\mathbb{E}_{(x_r,x_f)\sim(\mathbb{P},\mathbb{Q})}\left[f_1(C(x_r) - C(x_f))\right] - \mathbb{E}_{(x_r,x_f)\sim(\mathbb{P},\mathbb{Q})}\left[f_2(C(x_f) - C(x_r))\right]$$
$$= -\mathbb{E}_{(x_r,x_f)\sim(\mathbb{P},\mathbb{Q})}\left[f_1(C(x_r) - C(x_f))\right] - \mathbb{E}_{(x_r,x_f)\sim(\mathbb{P},\mathbb{Q})}\left[f_1(C(x_r) - C(x_f))\right]$$
$$= -2\mathbb{E}_{(x_r,x_f)\sim(\mathbb{P},\mathbb{Q})}\left[f_1(C(x_r) - C(x_f))\right].$$

If $g_1(y) = f_2(y)$ and $g_2(y) = f_1(y)$ (non-saturating GAN), we have that

$$L_G^{RGAN-NS} = \mathbb{E}_{(x_r,x_f)\sim(\mathbb{P},\mathbb{Q})}\left[g_1(C(x_r) - C(x_f))\right] + \mathbb{E}_{(x_r,x_f)\sim(\mathbb{P},\mathbb{Q})}\left[g_2(C(x_f) - C(x_r))\right]$$
$$= \mathbb{E}_{(x_r,x_f)\sim(\mathbb{P},\mathbb{Q})}\left[f_2(C(x_r) - C(x_f))\right] + \mathbb{E}_{(x_r,x_f)\sim(\mathbb{P},\mathbb{Q})}\left[f_1(C(x_f) - C(x_r))\right]$$
$$= \mathbb{E}_{(x_r,x_f)\sim(\mathbb{P},\mathbb{Q})}\left[f_1(C(x_f) - C(x_r))\right] + \mathbb{E}_{(x_r,x_f)\sim(\mathbb{P},\mathbb{Q})}\left[f_1(C(x_f) - C(x_r))\right]$$
$$= 2\mathbb{E}_{(x_r,x_f)\sim(\mathbb{P},\mathbb{Q})}\left[f_1(C(x_f) - C(x_r))\right].$$

## E  TESTING THE GRADIENT ARGUMENT

Previously, we argued that SGAN could be equivalent to IPM-GANs under very strict conditions and assumptions. We mentioned that although most assumptions are reasonable, the assumption that the generator is trained to optimality is unrealistic. In which case, SGAN would not be equivalent to IPM-based GANs since $D(x_r)$ would not reach 0.

As an experiment, we calculated the mini-batch average of $D(x_r)$ in the first 100 iterations of the training for the CAT dataset in 256x256. Note that SGAN becomes stuck at around 200 iterations and can never go beyond generating noise. Thus, a difference in the distribution of $D(x_r)$ could reveal something meaningful about why Relativistic GANs can converge while their non-relativistic counterparts cannot.

In those 100 iterations, we have that the distance between $\mathbb{P}$ and $\mathbb{Q}$ is maximal since $G$ only generate noise. Thus, we can perfectly distinguish real from fake data, which is one of the assumptions. The remaining assumptions were that $D$ and $G$ would be trained to optimality. Although we did not train $D$ more than once, after the discriminator step, we generally had that $D(x_r) \approx 1$. What we wanted to verify is whether $D(x_r) \approx 0$ after the generator step in Relativistic GANs, even though we did not train $G$ enough to reach optimality.

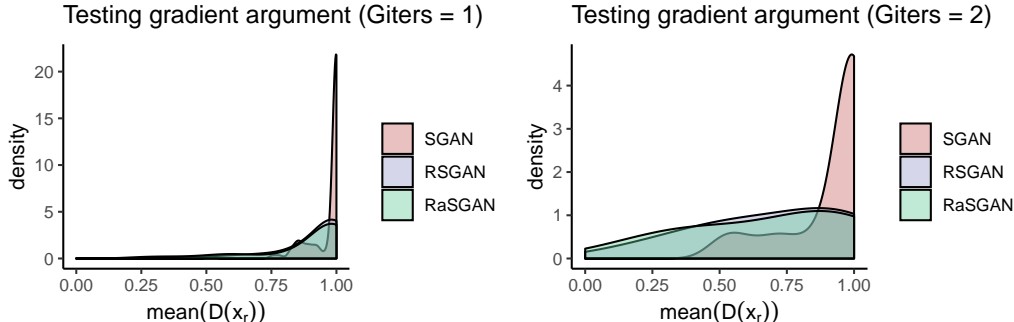

Figure 2: Density plots of the mini-batch average of $D(x_r)$ during the first 100 iterations of training on CAT with 256x256 images using only one or two generator updates per discriminator updates. If $D(x_r) = 0$ in all iterations, this would mean that the loss function would be the same as IPM-based GANs.

Results are shown in Figure 2. We observe that with only one generator update per discriminator update ($n_G = 1$), RSGAN and RaSGAN never reach an average $D(x_r)$ of 0 but the distribution is much less concentrated around 1 than with SGAN. With $n_G = 2$, RSGAN and RaSGAN sometimes reach an average $D(x_r)$ of 0 and they form an almost uniform distribution around $[0, 1]$. This suggests that with the missing property (i.e., using Relativistic GANs), SGAN can be made more similar to IPM-based GANs, but never equivalent. Thus, Relativistic Standard GANs can be seen as having a dynamic in-between SGAN and IPM-based GANs.

## F  CIFAR-10 HARD/UNSTABLE EXPERIMENTS

In these analyses, we compared SGAN, LSGAN, WGAN-GP, RSGAN, RaSGAN, RaLSGAN, and RaHingeGAN with the standard CNN architecture on unstable setups in CIFAR-10. Unless otherwise specified, we used $lr = .0002$, $\beta_1 = .5$, $\beta_2 = .999$, $n_D = 1$, and batch norm (Ioffe & Szegedy, 2015) in $G$ and $D$. We tested the following four unstable setups: (1) $lr = .001$, (2) $\beta_1 = .9$, $\beta_2 = .9$, (3) no batch norm in $G$ or $D$, and (4) all activation functions replaced with Tanh in both $G$ and $D$ (except for the output activation function of $D$).

Results are presented in Table 4. We observe that RaLSGAN performed better than LSGAN in all setups. RaHingeGAN performed slightly worse than HingeGAN in most setups. RSGAN and RaSGAN performed better than SGAN in two out of four setups, although differences were small. WGAN-GP generally performed poorly which we suspect is due to the single discriminator update per generator update. Overall, this provide good support for the improved stability of using the relative discriminator with LSGAN, but not with HingeGAN and SGAN. Although results are worse for the relativistic discriminator in some settings, differences are minimal and probably reflect natural variations.

It is surprising to observe low FID for SGAN without batch normalization considering its well-known difficulty with this setting (Arjovsky et al., 2017). Given these results, we suspected that CIFAR-10 may be too easy to fully observe the stabilizing effects of using the relative discriminator. Therefore, in the manuscript, we focused on the more difficult CAT dataset with high resolution pictures.

## G  LOSS FUNCTIONS USED IN EXPERIMENTS

### G.1  SGAN (NON-SATURATING)

$$L_D^{SGAN} = -\mathbb{E}_{x_r \sim \mathbb{P}}\left[\log\left(\text{sigmoid}(C(x_r))\right)\right] - \mathbb{E}_{x_f \sim \mathbb{Q}}\left[\log\left(1 - \text{sigmoid}(C(x_f))\right)\right] \quad (20)$$

$$L_G^{SGAN} = -\mathbb{E}_{x_f \sim \mathbb{Q}}\left[\log\left(\text{sigmoid}(C(x_f))\right)\right] \quad (21)$$

### G.2  RSGAN

$$L_D^{RSGAN} = -\mathbb{E}_{(x_r, x_f) \sim (\mathbb{P}, \mathbb{Q})}\left[\log(\text{sigmoid}(C(x_r) - C(x_f)))\right] \quad (22)$$

Table 4: Fréchet Inception Distance (FID) at exactly 100k generator iterations on the CIFAR-10 dataset using unstable setups with different GAN loss functions. Unless otherwise specified, we used $lr = .0002$, $\beta = (.50, .999)$, $n_D = 1$, and batch norm (BN) in $D$ and $G$. All models were trained using the same *a priori* selected seed (seed=1).

| Loss | $lr = .001$ | $\beta = (.9, .9)$ | No BN | Tanh |
|---|---|---|---|---|
| SGAN | 154.20 | 35.29 | 35.54 | 59.17 |
| RSGAN | 50.95 | 45.12 | 37.11 | 77.21 |
| RaSGAN | 55.55 | 43.46 | 41.96 | 54.42 |
| LSGAN | 52.27 | 225.94 | 38.54 | 147.87 |
| RaLSGAN | **33.33** | 48.92 | **34.66** | 53.07 |
| HingeGAN | 43.28 | **33.47** | 34.21 | 58.51 |
| RaHingeGAN | 51.05 | 42.78 | 43.75 | **50.69** |
| WGAN-GP | 61.97 | 104.95 | 85.27 | 59.94 |

$$L_G^{RSGAN} = -\mathbb{E}_{(x_r, x_f) \sim (\mathbb{P}, \mathbb{Q})} \left[ \log(\text{sigmoid}(C(x_f) - C(x_r))) \right] \tag{23}$$

### G.3 RASGAN

$$L_D^{RaSGAN} = -\mathbb{E}_{x_r \sim \mathbb{P}} \left[ \log \left( \tilde{D}(x_r) \right) \right] - \mathbb{E}_{x_f \sim \mathbb{Q}} \left[ \log \left( 1 - \tilde{D}(x_f) \right) \right] \tag{24}$$

$$L_G^{RaSGAN} = -\mathbb{E}_{x_f \sim \mathbb{Q}} \left[ \log \left( \tilde{D}(x_f) \right) \right] - \mathbb{E}_{x_r \sim \mathbb{P}} \left[ \log \left( 1 - \tilde{D}(x_r) \right) \right] \tag{25}$$

$\tilde{D}(x_r) = \text{sigmoid} \left( C(x_r) - \mathbb{E}_{x_f \sim \mathbb{Q}} C(x_f) \right)$
$\tilde{D}(x_f) = \text{sigmoid} \left( C(x_f) - \mathbb{E}_{x_r \sim \mathbb{P}} C(x_r) \right)$

### G.4 LSGAN

$$L_D^{LSGAN} = \mathbb{E}_{x_r \sim \mathbb{P}} \left[ (C(x_r) - 0)^2 \right] + \mathbb{E}_{x_f \sim \mathbb{Q}} \left[ (C(x_f) - 1)^2 \right] \tag{26}$$

$$L_G^{LSGAN} = \mathbb{E}_{x_f \sim \mathbb{Q}} \left[ (C(x_f) - 0)^2 \right] \tag{27}$$

### G.5 RALSGAN

$$L_D^{RaLSGAN} = \mathbb{E}_{x_r \sim \mathbb{P}} \left[ (C(x_r) - \mathbb{E}_{x_f \sim \mathbb{Q}} C(x_f) - 1)^2 \right] + \mathbb{E}_{x_f \sim \mathbb{Q}} \left[ (C(x_f) - \mathbb{E}_{x_r \sim \mathbb{P}} C(x_r) + 1)^2 \right] \tag{28}$$

$$L_G^{RaLSGAN} = \mathbb{E}_{x_f \sim \mathbb{P}} \left[ (C(x_f) - \mathbb{E}_{x_r \sim \mathbb{P}} C(x_r) - 1)^2 \right] + \mathbb{E}_{x_r \sim \mathbb{P}} \left[ (C(x_r) - \mathbb{E}_{x_f \sim \mathbb{Q}} C(x_f) + 1)^2 \right] \tag{29}$$

### G.6 HINGEGAN

$$L_D^{HingeGAN} = \mathbb{E}_{x_r \sim \mathbb{P}} \left[ \max(0, 1 - C(x_r)) \right] + \mathbb{E}_{x_f \sim \mathbb{Q}} \left[ \max(0, 1 + C(x_f)) \right] \tag{30}$$

$$L_G^{HingeGAN} = -\mathbb{E}_{x_f \sim \mathbb{Q}} \left[ C(x_f) \right] \tag{31}$$

### G.7 RAHINGEGAN

$$L_D^{HingeGAN} = \mathbb{E}_{x_r \sim \mathbb{P}} \left[ \max(0, 1 - \tilde{D}(x_r)) \right] + \mathbb{E}_{x_f \sim \mathbb{Q}} \left[ \max(0, 1 + \tilde{D}(x_f)) \right] \tag{32}$$

$$L_G^{HingeGAN} = \mathbb{E}_{x_f \sim \mathbb{P}} \left[ \max(0, 1 - \tilde{D}(x_f)) \right] + \mathbb{E}_{x_r \sim \mathbb{Q}} \left[ \max(0, 1 + \tilde{D}(x_r)) \right] \tag{33}$$

$\tilde{D}(x_r) = C(x_r) - \mathbb{E}_{x_f \sim \mathbb{Q}} C(x_f)$
$\tilde{D}(x_f) = C(x_f) - \mathbb{E}_{x_r \sim \mathbb{P}} C(x_r)$

## G.8 WGAN-GP

$$L_D^{WGAN-GP} = -\mathbb{E}_{x_r \sim \mathbb{P}}\left[C(x_r)\right] + \mathbb{E}_{x_f \sim \mathbb{Q}}\left[C(x_f)\right] + \lambda\mathbb{E}_{\hat{x} \sim \mathbb{P}_{\hat{x}}}\left[\left(||\nabla_{\hat{x}}C(\hat{x})||_2 - 1\right)^2\right] \tag{34}$$

$$L_G^{WGAN-GP} = -\mathbb{E}_{x_f \sim \mathbb{Q}}\left[C(x_f)\right] \tag{35}$$

$\mathbb{P}_{\hat{x}}$ is the distribution of $\hat{x} = \epsilon x_r + (1 - \epsilon)x_f$, where $x_r \sim \mathbb{P}$, $x_f \sim \mathbb{Q}$, $\epsilon \sim U[0, 1]$.

## G.9 RSGAN-GP

$$L_D^{RSGAN} = -\mathbb{E}_{(x_r, x_f) \sim (\mathbb{P}, \mathbb{Q})}\left[\log(\text{sigmoid}(C(x_r) - C(x_f)))\right] + \lambda\mathbb{E}_{\hat{x} \sim \mathbb{P}_{\hat{x}}}\left[\left(||\nabla_{\hat{x}}C(\hat{x})||_2 - 1\right)^2\right] \tag{36}$$

$$L_G^{RSGAN} = -\mathbb{E}_{(x_r, x_f) \sim (\mathbb{P}, \mathbb{Q})}\left[\log(\text{sigmoid}(C(x_f) - C(x_r)))\right] \tag{37}$$

$\mathbb{P}_{\hat{x}}$ is the distribution of $\hat{x} = \epsilon x_r + (1 - \epsilon)x_f$, where $x_r \sim \mathbb{P}$, $x_f \sim \mathbb{Q}$, $\epsilon \sim U[0, 1]$.

## G.10 RASGAN-GP

$$L_D^{RaSGAN} = -\mathbb{E}_{x_r \sim \mathbb{P}}\left[\log\left(\tilde{D}(x_r)\right)\right] - \mathbb{E}_{x_f \sim \mathbb{Q}}\left[\log\left(1 - \tilde{D}(x_f)\right)\right] + \lambda\mathbb{E}_{\hat{x} \sim \mathbb{P}_{\hat{x}}}\left[\left(||\nabla_{\hat{x}}C(\hat{x})||_2 - 1\right)^2\right] \tag{38}$$

$$L_G^{RaSGAN} = -\mathbb{E}_{x_f \sim \mathbb{Q}}\left[\log\left(\tilde{D}(x_f)\right)\right] - \mathbb{E}_{x_r \sim \mathbb{P}}\left[\log\left(1 - \tilde{D}(x_r)\right)\right] \tag{39}$$

$\tilde{D}(x_r) = \text{sigmoid}\left(C(x_r) - \mathbb{E}_{x_f \sim \mathbb{Q}}C(x_f)\right)$
$\tilde{D}(x_f) = \text{sigmoid}\left(C(x_f) - \mathbb{E}_{x_r \sim \mathbb{P}}C(x_r)\right)$
$\mathbb{P}_{\hat{x}}$ is the distribution of $\hat{x} = \epsilon x_r + (1 - \epsilon)x_f$, where $x_r \sim \mathbb{P}$, $x_f \sim \mathbb{Q}$, $\epsilon \sim U[0, 1]$.

## H ALGORITHMS

---
**Algorithm 1** Training algorithm for non-saturating RGANs with symmetric loss functions

---
**Require:** The number of $D$ iterations $n_D$ ($n_D = 1$ unless one seeks to train $D$ to optimality), batch size $m$, and functions $f$ which determine the objective function of the discriminator ($f$ is $f_1$ from equation 10 assuming that $f_2(-y) = f_1(y)$, which is true for many GANs).
    **while** $\theta$ has not converged **do**
        **for** $t = 1, \ldots, n_D$ **do**
            Sample $\{x^{(i)}\}_{i=1}^m \sim \mathbb{P}$
            Sample $\{z^{(i)}\}_{i=1}^m \sim \mathbb{P}_z$
            Update $w$ using SGD by ascending with $\nabla_w \frac{1}{m}\sum_{i=1}^m \left[f(C_w(x^{(i)}) - C_w(G_\theta(z^{(i)})))\right]$
        **end for**
        Sample $\{x^{(i)}\}_{i=1}^m \sim \mathbb{P}$
        Sample $\{z^{(i)}\}_{i=1}^m \sim \mathbb{P}_z$
        Update $\theta$ using SGD by ascending with $\nabla_\theta \frac{1}{m}\sum_{i=1}^m \left[f(C_w(G_\theta(z^{(i)})) - C_w(x^{(i)}))\right]$
    **end while**

---

---

**Algorithm 2** Training algorithm for non-saturating RaGANs

---

**Require:** The number of $D$ iterations $n_D$ ($n_D = 1$ unless one seek to train $D$ to optimality), batch size $m$, and functions $f_1$ and $f_2$ which determine the objective function of the discriminator (see equation 10).

    **while** $\theta$ has not converged **do**
        **for** $t = 1, \ldots, n_D$ **do**
            Sample $\{x^{(i)}\}_{i=1}^{m} \sim \mathbb{P}$
            Sample $\{z^{(i)}\}_{i=1}^{m} \sim \mathbb{P}_z$
            Let $\overline{C_w(x_r)} = \frac{1}{m} \sum_{i=1}^{m} C_w(x^{(i)})$
            Let $\overline{C_w(x_f)} = \frac{1}{m} \sum_{i=1}^{m} C_w(G_\theta(z^{(i)}))$
            Update $w$ using SGD by ascending with

$$\nabla_w \frac{1}{m} \sum_{i=1}^{m} \left[ f_1(C_w(x^{(i)}) - \overline{C_w(x_f)}) + f_2(C_w(G_\theta(z^{(i)})) - \overline{C_w(x_r)}) \right]$$

        **end for**
        Sample $\{x^{(i)}\}_{i=1}^{m} \sim \mathbb{P}$
        Sample $\{z^{(i)}\}_{i=1}^{m} \sim \mathbb{P}_z$
        Let $\overline{C_w(x_r)} = \frac{1}{m} \sum_{i=1}^{m} C_w(x^{(i)})$
        Let $\overline{C_w(x_f)} = \frac{1}{m} \sum_{i=1}^{m} C_w(G_\theta(z^{(i)}))$
        Update $\theta$ using SGD by ascending with

$$\nabla_\theta \frac{1}{m} \sum_{i=1}^{m} \left[ f_1(C_w(G_\theta(z^{(i)})) - \overline{C_w(x_r)}) + f_2(C_w(x^{(i)}) - \overline{C_w(x_f)}) \right]$$

    **end while**

---

## I   ARCHITECTURES

### I.1   STANDARD CNN

| Generator |
|---|
| $z \in \mathbb{R}^{128} \sim N(0, I)$ |
| linear, 128 -> 512*4*4 |
| Reshape, 512*4*4 -> 512 x 4 x 4 |
| ConvTranspose2d 4x4, stride 2, pad 1, 512->256 |
| BN and ReLU |
| ConvTranspose2d 4x4, stride 2, pad 1, 256->128 |
| BN and ReLU |
| ConvTranspose2d 4x4, stride 2, pad 1, 128->64 |
| BN and ReLU |
| ConvTranspose2d 3x3, stride 1, pad 1, 64->3 |
| Tanh |

| Discriminator |
|---|
| $x \in \mathbb{R}^{3x32x32}$ |
| Conv2d 3x3, stride 1, pad 1, 3->64 |
| LeakyReLU 0.1 |
| Conv2d 4x4, stride 2, pad 1, 64->64 |
| LeakyReLU 0.1 |
| Conv2d 3x3, stride 1, pad 1, 64->128 |
| LeakyReLU 0.1 |
| Conv2d 4x4, stride 2, pad 1, 128->128 |
| LeakyReLU 0.1 |
| Conv2d 3x3, stride 1, pad 1, 128->256 |
| LeakyReLU 0.1 |
| Conv2d 4x4, stride 2, pad 1, 256->256 |
| LeakyReLU 0.1 |
| Conv2d 3x3, stride 1, pad 1, 256->512 |
| Reshape, 512 x 4 x 4 -> 512*4*4 |
| linear, 512*4*4 -> 1 |

## I.2 DCGAN 64x64

| Generator |
|---|
| $z \in \mathbb{R}^{128} \sim N(0, I)$ |
| ConvTranspose2d 4x4, stride 1, pad 0, no bias, 128->512 |
| BN and ReLU |
| ConvTranspose2d 4x4, stride 2, pad 1, no bias, 512->256 |
| BN and ReLU |
| ConvTranspose2d 4x4, stride 2, pad 1, no bias, 256->128 |
| BN and ReLU |
| ConvTranspose2d 4x4, stride 2, pad 1, no bias, 128->64 |
| BN and ReLU |
| ConvTranspose2d 4x4, stride 2, pad 1, no bias, 64->3 |
| Tanh |

| Discriminator |
|---|
| $x \in \mathbb{R}^{3x64x64}$ |
| Conv2d 4x4, stride 2, pad 1, no bias, 3->64 |
| LeakyReLU 0.2 |
| Conv2d 4x4, stride 2, pad 1, no bias, 64->128 |
| BN and LeakyReLU 0.2 |
| Conv2d 4x4, stride 2, pad 1, no bias, 128->256 |
| BN and LeakyReLU 0.2 |
| Conv2d 4x4, stride 2, pad 1, no bias, 256->512 |
| BN and LeakyReLU 0.2 |
| Conv2d 4x4, stride 2, pad 1, no bias, 512->1 |

## I.3 DCGAN 128x128

| Generator |
|---|
| $z \in \mathbb{R}^{128} \sim N(0, I)$ |
| ConvTranspose2d 4x4, stride 1, pad 0, no bias, 128->1024 |
| BN and ReLU |
| ConvTranspose2d 4x4, stride 2, pad 1, no bias, 1024->512 |
| BN and ReLU |
| ConvTranspose2d 4x4, stride 2, pad 1, no bias, 512->256 |
| BN and ReLU |
| ConvTranspose2d 4x4, stride 2, pad 1, no bias, 256->128 |
| BN and ReLU |
| ConvTranspose2d 4x4, stride 2, pad 1, no bias, 128->64 |
| BN and ReLU |
| ConvTranspose2d 4x4, stride 2, pad 1, no bias, 64->3 |
| Tanh |

| Discriminator |
|---|
| $x \in \mathbb{R}^{3x128x128}$ |
| Conv2d 4x4, stride 2, pad 1, no bias, 3->64 |
| LeakyReLU 0.2 |
| Conv2d 4x4, stride 2, pad 1, no bias, 64->128 |
| BN and LeakyReLU 0.2 |
| Conv2d 4x4, stride 2, pad 1, no bias, 128->256 |
| BN and LeakyReLU 0.2 |
| Conv2d 4x4, stride 2, pad 1, no bias, 256->512 |
| BN and LeakyReLU 0.2 |
| Conv2d 4x4, stride 2, pad 1, no bias, 512->1024 |
| BN and LeakyReLU 0.2 |
| Conv2d 4x4, stride 2, pad 1, no bias, 1024->1 |

### I.4 DCGAN 256x256

| Generator | Discriminator (PACGAN2 (Lin et al., 2017)) |
|---|---|
| $z \in \mathbb{R}^{128} \sim N(0, I)$ | $x_1 \in \mathbb{R}^{3x256x256}, x_2 \in \mathbb{R}^{3x256x256}$ |
| ConvTranspose2d 4x4, stride 1, pad 0, no bias, 128->1024 | Concatenate $[x_1, x_2] \in \mathbb{R}^{6x256x256}$ |
| BN and ReLU | Conv2d 4x4, stride 2, pad 1, no bias, 6->32 |
| ConvTranspose2d 4x4, stride 2, pad 1, no bias, 1024->512 | LeakyReLU 0.2 |
| BN and ReLU | Conv2d 4x4, stride 2, pad 1, no bias, 32->64 |
| ConvTranspose2d 4x4, stride 2, pad 1, no bias, 512->256 | LeakyReLU 0.2 |
| BN and ReLU | Conv2d 4x4, stride 2, pad 1, no bias, 64->128 |
| ConvTranspose2d 4x4, stride 2, pad 1, no bias, 256->128 | BN and LeakyReLU 0.2 |
| BN and ReLU | Conv2d 4x4, stride 2, pad 1, no bias, 128->256 |
| ConvTranspose2d 4x4, stride 2, pad 1, no bias, 128->64 | BN and LeakyReLU 0.2 |
| BN and ReLU | Conv2d 4x4, stride 2, pad 1, no bias, 256->512 |
| ConvTranspose2d 4x4, stride 2, pad 1, no bias, 64->32 | BN and LeakyReLU 0.2 |
| BN and ReLU | Conv2d 4x4, stride 2, pad 1, no bias, 512->1024 |
| ConvTranspose2d 4x4, stride 2, pad 1, no bias, 64->3 | BN and LeakyReLU 0.2 |
| Tanh | Conv2d 4x4, stride 2, pad 1, no bias, 1024->1 |

## J SAMPLES

This shows a selection of cats from certain models. Images shown are from the lowest FID registered at every 10k generator iterations. Given space constraint, with cats in high resolution, we show some of the nicer looking cats for each approach, there are evidently some worse looking cats (See `https://github.com/AlexiaJM/RelativisticGAN/tree/master/images/full_minibatch` for all cats of the mini-batch).

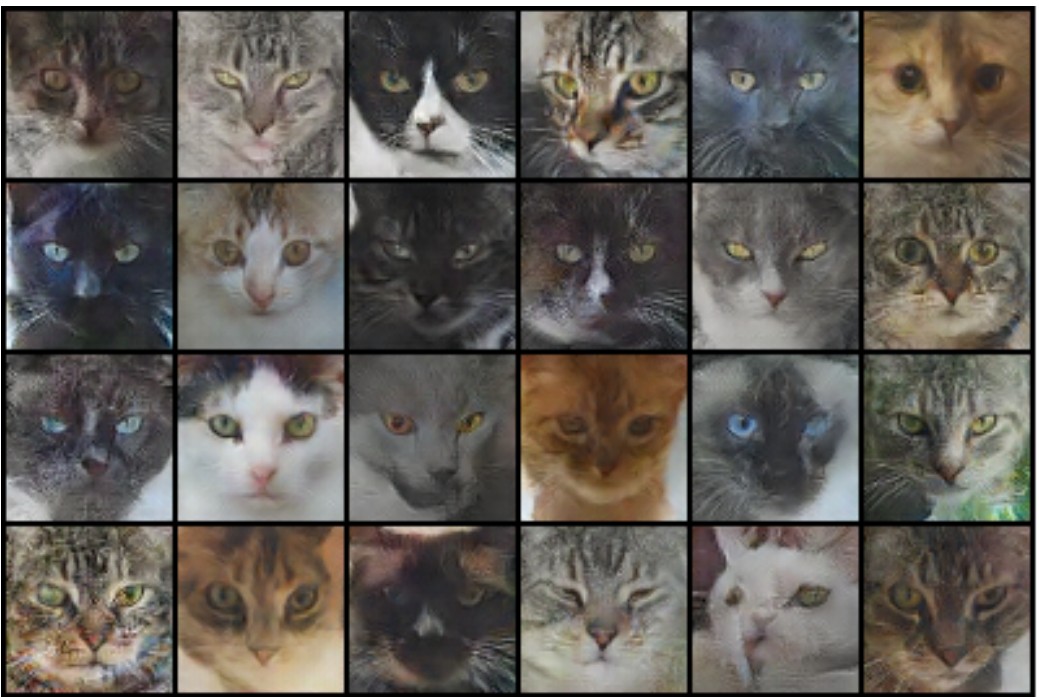

Figure 3: 64x64 cats with RaLSGAN (FID = 11.97)

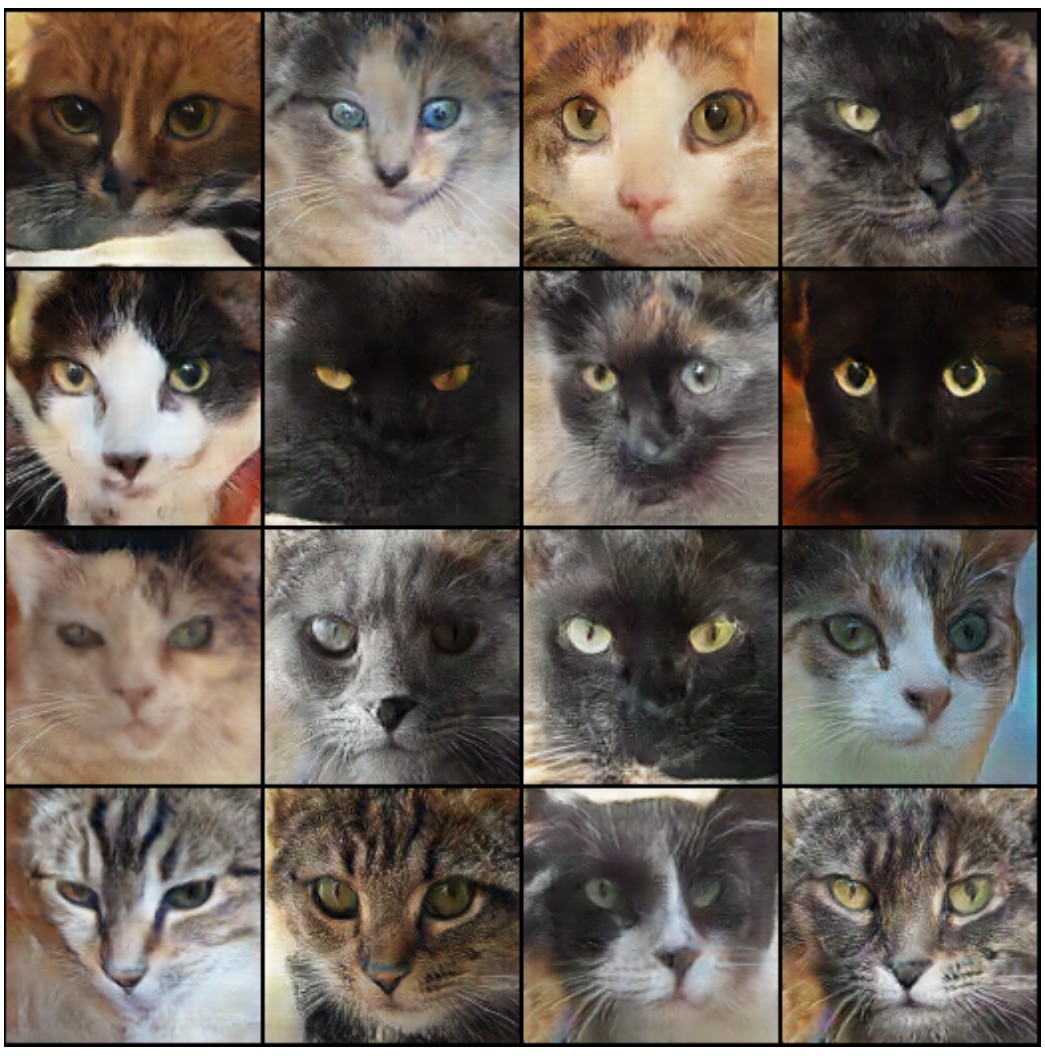

Figure 4: 128x128 cats with RaLSGAN (FID = 15.85)

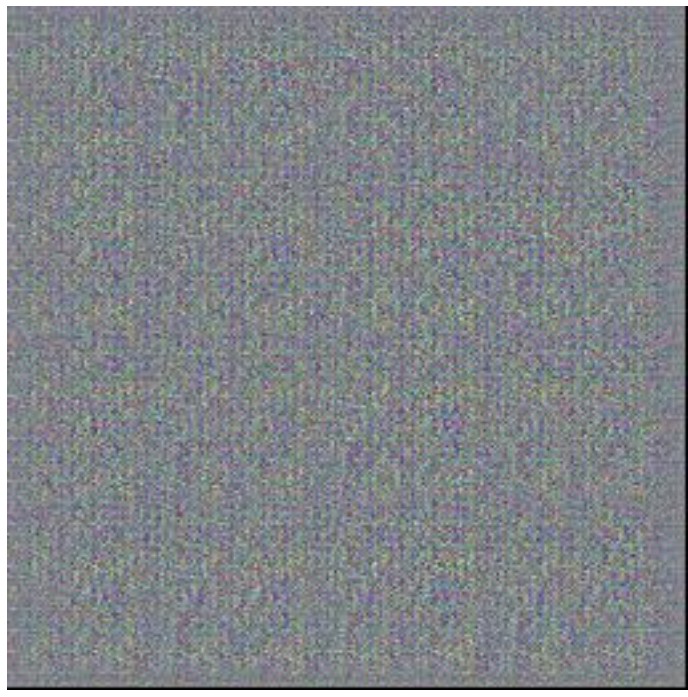

Figure 5: 256x256 cats with GAN (5k iterations)

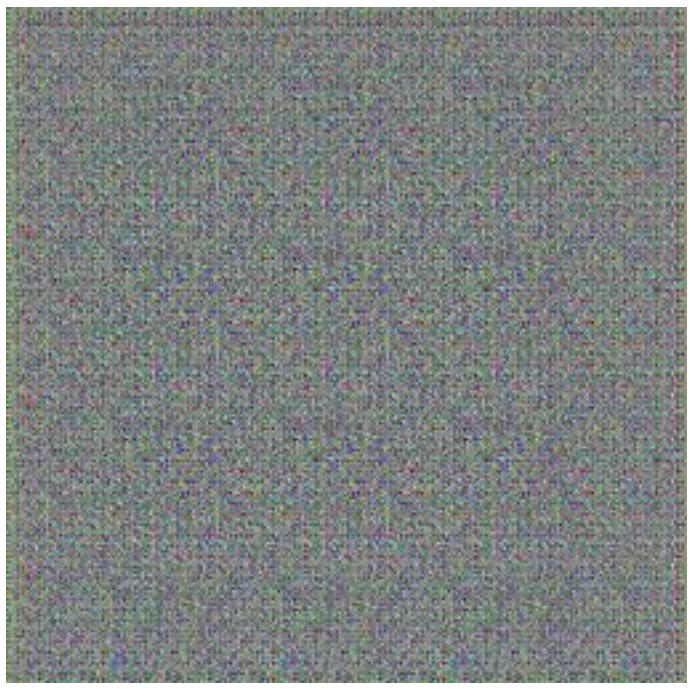

Figure 6: 256x256 cats with LSGAN (5k iterations)

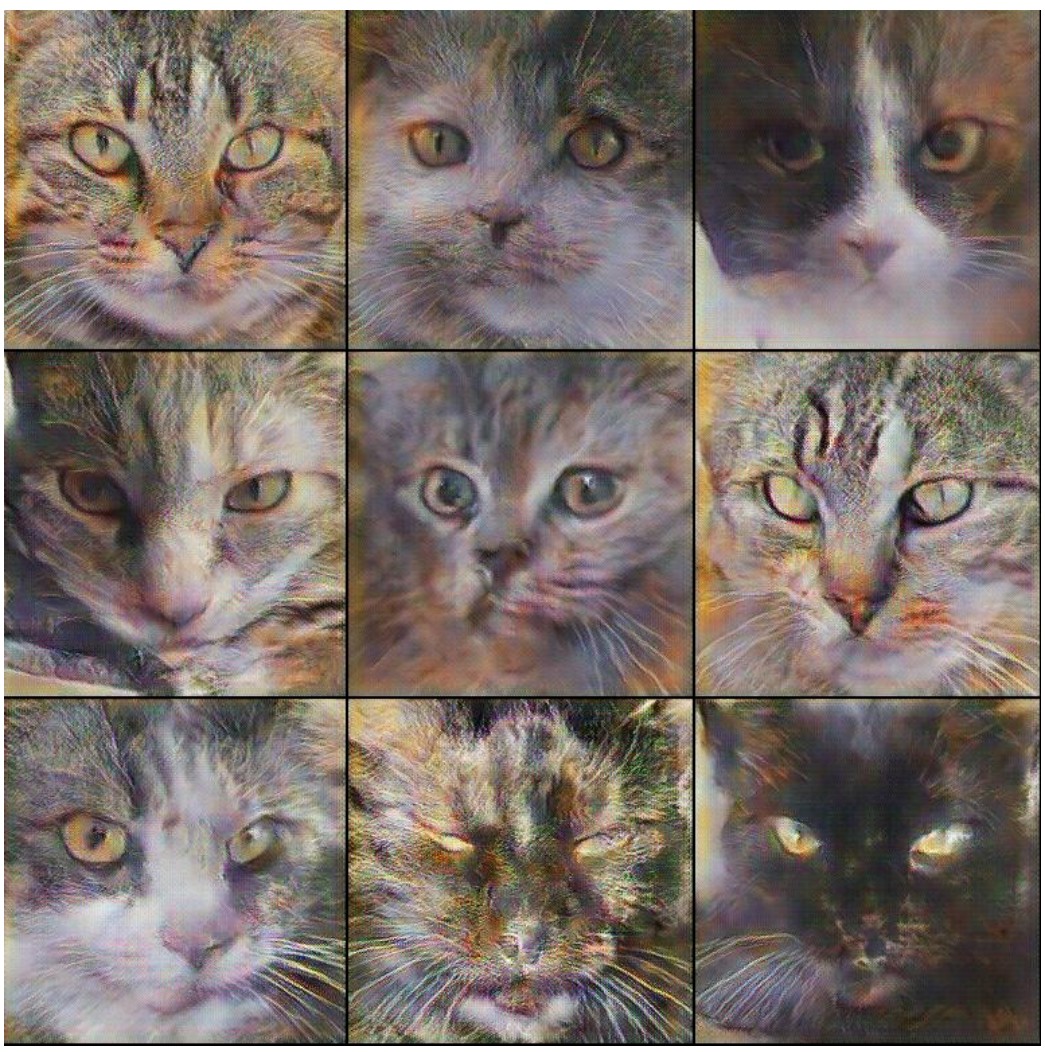

Figure 7: 256x256 cats with RaSGAN (FID = 32.11)

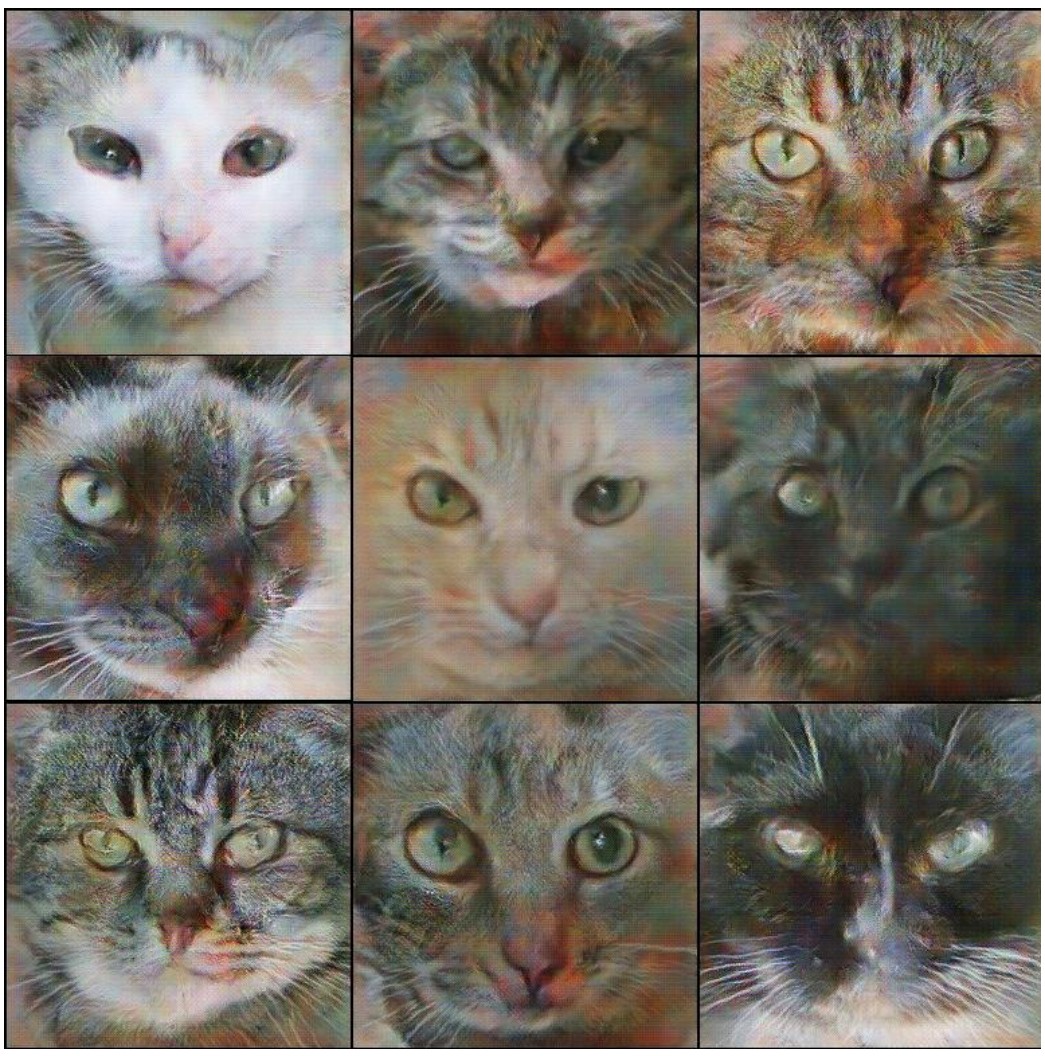

Figure 8: 256x256 cats with RaLSGAN (FID = 35.21)

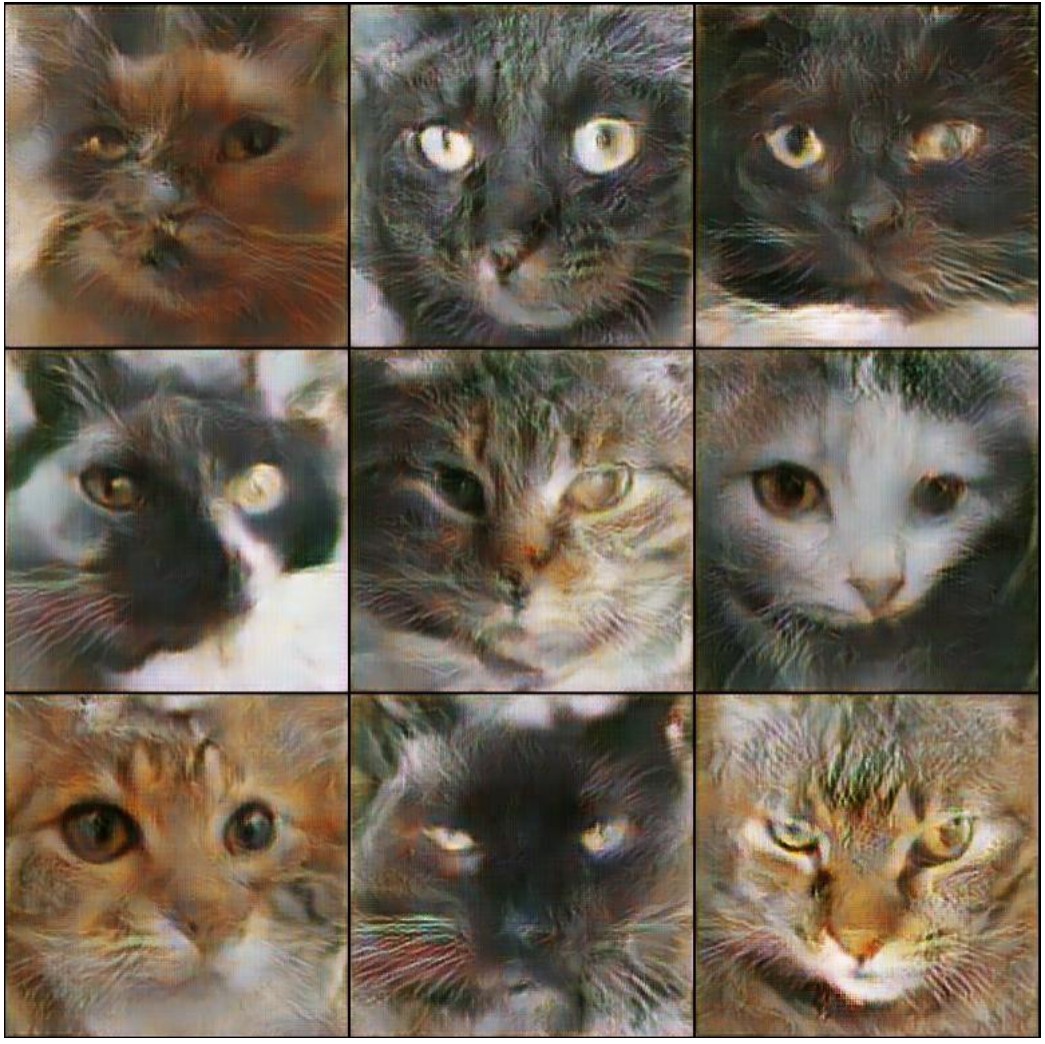

Figure 9: 256x256 cats with SpectralSGAN (FID = 54.73)

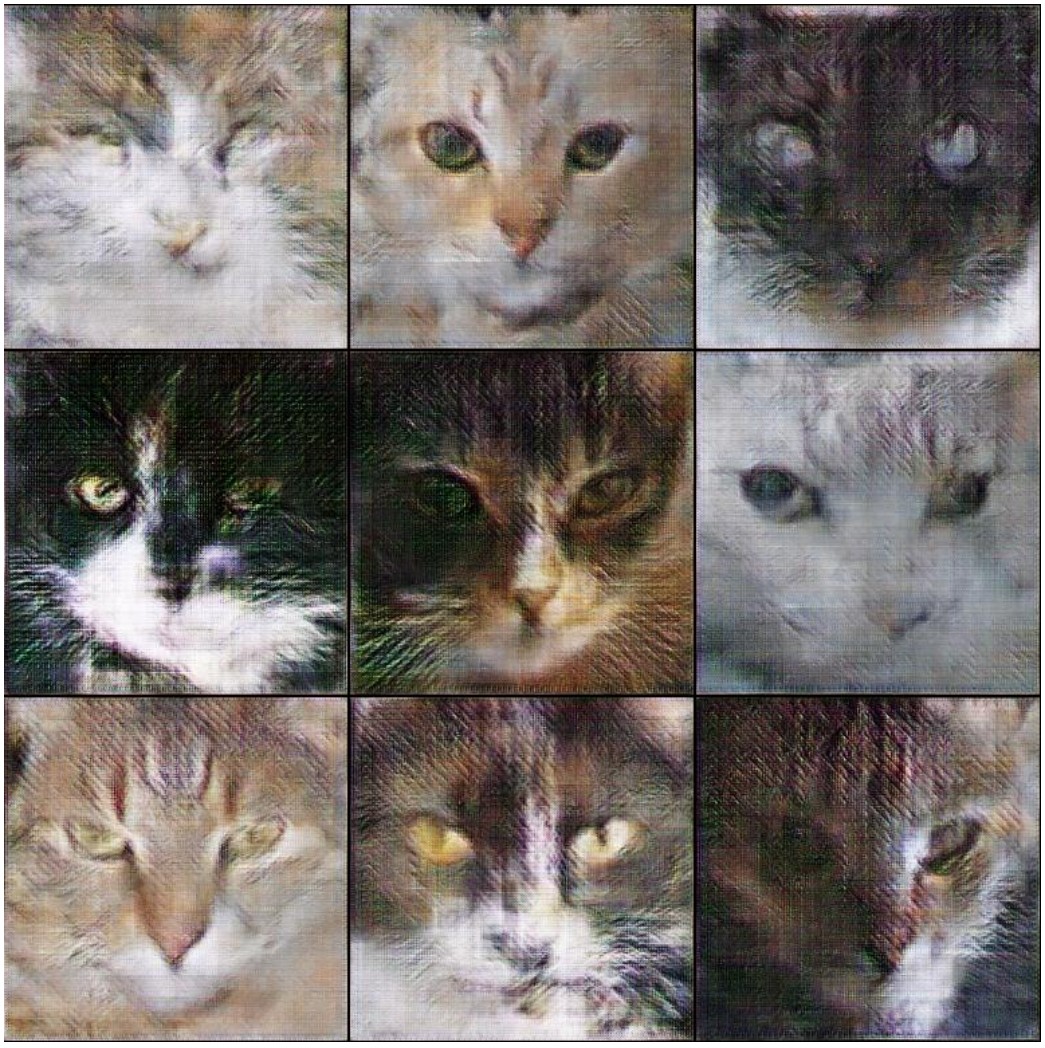

Figure 10: 256x256 cats with WGAN-GP (FID > 100)

