# OpenReview forum: " The relativistic discriminator: a key element missing from standard GAN"
_ICLR.cc/2019/Conference_

### Official Review · AnonReviewer3 · 2018-11-02
**Review: The relativistic Discriminator: A key element missing from standard GAN**

**Rating:** 7
**Confidence:** 3

**Review:**


In this work, the authors considers a variation of GAN by consider simultaneously decrease the probability that real data is real for the generator. To include such a property, the authors propose a relativistic discriminator which estimate the probability that the given real data is more realistic than the fake data. Numerical results are performed to show that the proposed methods are effective, and the resulting GANs are relatively more stable and generate higher quality data samples than their non-relativistic counterparts.

Overall the paper is well written and the rationale behind the proposed modification is clear. In particular, the authors use three different perspective, (the prior knowledge, the divergence minimization, and the gradient expressions), to explain what they thought is missing in the state-of-the-art. By proposing to utilize the information about both real and fake data in the discriminator definition, the authors’ have (to some extent) alleviated the above shortcoming of the state-of-the-art.  Unfortunately, like almost all papers related to the field,  there has been no rigorously justification behind the proposed methods.

The English of the paper has to be significantly improved. For example, grammar errors like “this mean….”, “didn’t converge, …”

Unfortunately, the codes of the paper is not released, I will encourage the authors to do so.

---

> ### Author Response · Authors · 2018-11-13
> **Review 3**
>
> Dear Reviewer 3,
>
> Thank you for your comments.
>
> We reviewed the paper to correct for spelling mistakes and to make it less familiar (by removing contractions). According to Reviewer 1 suggestions, we also revised Section 3 to improve the wording and explanations.
>
> The code has already been released through GitHub. To retain anonymity, we re-uploaded the GitHub repository without any information relating to the authors: https://github.com/anonymousconference/RGAN.

---

### Official Review · AnonReviewer2 · 2018-11-03
**Interesting idea**

**Rating:** 6
**Confidence:** 4

**Review:**

The paper proposes a “relativistic discriminator” which has the property that the probability of real data being real decreases as the probability of fake data being real increases.

The paper is very well-written. I particularly liked Section 3 which motivates the key idea through multiple viewpoints. The experiments show that the relativistic discriminator helps in some settings, although it does seem a bit sensitive to hyperparameters, architectures and datasets.

I found the argument about connections to IPM-GANs a bit confusing. In a couple of places in Section 4, the relativistic loss is motivated by showing that the relativistic discriminator makes SGANs more like IPM-GANs. However, not all IPM-GANs are the same, e.g. the experiments show performance gaps between RSGAN, RaSGAN, and WGAN-GP, which suggests there could be other confounding factors.

Could you devise experiments on synthetic datasets where the different hypotheses in Section 3 might lead to different solutions? Would be very interesting to see which hypothesis best explains why relativistic discriminator helps!

Section 4.3: How do you justify the averaging? While the relativistic GAN is well-explained, section 4.3 only briefly mentions the averaging idea. Given that averaging seems to help a lot in some of the experiments, it’d be great to see further discussion of why this helps.

---

> ### Author Response · Authors · 2018-11-13
> **Reviewer 2**
>
> Dear Reviewer 2,
>
> Thank you for your comments.
>
> We are in agreement about the fact that IPM-based GANs are different from Relativistic GANs. They are similar, but yet different enough that they are not of the same class. Although our paper mentioned the similarity, it did not mention the difference which could lead to readers thinking that IPM-Based GANs are a subset of Relativistic GANs (and we talked to people who thought this was the case after reading our paper). In section 4.2 p5, we now highlight better the differences and similarities:
> “If one use the identity function (i.e., f_1(y)=g_2(y)=-y, f_2(y)=g_1(y)=y), this results in a degenerate case since there is no supremum/maximum. However, if one adds a constraint so that C(x_r)-C(x_f) is bounded, then there is a supremum and one arrives at IPM-based GANs. Thus, although different, IPM-based GANs share a very similar loss function focused on the difference in critics.”
>
> As you suggested, we ran some additional experiments focused on testing the gradient argument (see Appendix E, p13-14). Although the gradient argument applies if we train G to optimality; in practice, we do not train G to optimality. Thus, we observed that RSGAN/RaSGAN are not equivalent to IPM-based GANs in real-world scenarios. However, they act in a way that is somewhere in-between the dynamics of SGAN and IPM-based GANs. In addition to Appendix E, we now also mention that training G to optimality is an unrealistic assumption in Section 3.3 p4.
>
> The main intuition that led to Relativistic average GANs was actually in our initial paper version, but it was removed due to space constraints (8 pages max). Given your comment, we decided to relay it to the Appendix rather than completely removing it; it is now in Appendix B p11-12. Additionally, we added the following sentence at the beginning of section 4.3 p5:
> “The discriminator has a very different interpretation in SGAN compared to RSGAN. In SGAN, D(x) estimates the probability that x is real, while in RGANs, D(x_r,x_f) estimates the probability that x_r is more realistic than x_f. As a middle ground, we developed an alternative to the Relativistic Discriminator, which retains approximately the same interpretation as the discriminator in SGAN while still being relativistic.”
>
> This explains why we created RaGANs, but it does not explain why they generally perform better than RGANs. We are still uncertain as to why RGANs perform less well than RaGANs, given that both approaches improve stability.

---

### Official Review · AnonReviewer1 · 2018-11-03
**Tweak on the Standard GAN**

**Rating:** 6
**Confidence:** 2

**Review:**

The paper describes an interesting tweak of the standard GAN model (inspired by IPM based GANs) where both the generator and the discriminator optimize relative realness (and fakeness) of the (real, fake) image pairs. The authors give some intuition for this tweak and ran experiments with CIFAR10 and CAT datasets. Different variants of the standard GAN and the new tweak were compared under the FID metric. The experimental setup and details are provided; and the code is made publicly available.

The results are good and their tweak seems to help in most of the cases. The paper, however, is not very well written and is not of publication quality.  All the insights given in Section 3 are wrong, incomplete and unsatisfying. For example, in Section 3.4, the authors suggest that gradient dynamics of the tweaked model (with some unrealistic and infeasible assumptions) is same as that of an IPM-GAN and contribute to stability. This is wrong. Similar dynamics (even under the unrealistic assumption), does not imply similar performance. In fact, if one is trying to move towards IPM dynamics, then one should try to tweak an IPM model directly. Section 3.2 also seems wrong from my understanding of GAN training. Section 3.3 could also be improved. In fact, any explanations based on minimizing JS divergence is incomplete without answering as to why JS divergence minimizing is the best thing to do.

The author should have provided more comparison images to rule out the fact that the tweak is not overfitting for the FID metric. The benchmarks are also weak and more experiments need to be done (Eg, CelebA).

---

> ### Author Response · Authors · 2018-11-14
> **Review 1**
>
> Dear Reviewer 1,
>
> Thank you for your comments.
>
> We hope that this message will find you well. We really took the time to review all your comments and in doing so we significantly improved the paper. As you suggested, one aspect (the gradient argument) was relying on unrealistic assumptions (that G would be trained to optimality). We believe that we were able to make the paper of much higher quality so please consider this response in your assessment of the paper.
>
> You mention that the paper is “not well written” and a lot of your emphasis is on Section 3. To remedy your concerns, we spent a lot of time to rewrite parts of it in a way that is much clearer. Also, as suggested by Reviewer 3, we reviewed corrected spelling mistakes and removed contractions to make it less familiar.
>
> Note that we removed section 3.1 since it was not a real subsection.
>
> Regarding Section 3.2 (which is now section 3.1), we rewrote it because it was somewhat unclear after we removed so much text to fit the 8 pages limit.
>
> Regarding Section (3.3, which is now section 3.2), we clarified that JSD is not the only divergence where we see something like Figure 1a, this is true for most divergences. Thus, our explanation is not incomplete. See below:
> “Note that although specific to the JSD, similar dynamics are true for other divergences; when the divergence is maximal, D(x_r) and D(x_f) are very far from one another, but they converge to the same value as the divergence approach zero. Thus, this argument applies to other divergences.”
>
> Regarding Section (3.4, which is now section 3.3), we agree that one assumption was unrealistic. The problematic assumption was assuming that both D and G are trained to optimality. In practice, certain GANs (mostly IPM-based GANs) train D multiple times. However, no GANs to our knowledge train G multiple times since GANs do not converge when doing so. G can only take a small step at a time; otherwise, the generator will collapse early on. Note that Reviewer 3 suggested that we do some experiments regarding the gradient argument and we did (the full experiment described below is in Appendix E). We observed that we do not reach D(x_r)=0 using relativistic GANs when n_G = 1 (the number of generator update per critic’s updates). If using n_G = 2, it does sometimes happen that D(x_r)=0. Either way, we have that RSGAN significantly increase the proportion of low D(x_r) even if it rarely reaches 0. Thus, although we cannot make SGAN equivalent to IPM-based GANs, we can make them more similar. We rectify this in p4.
>
> To respond to your comments about IPMs, we seek to find a GAN with a similar dynamic to IPM-based GANs without actually using IPMs. We want this because IPM-based GANs have an important drawback: they tend to be very computationally demanding (not always, but more often than not). In the introduction, we now mention that IPM-based GANs tend to be longer to train. Thus, finding an approach with similar stability, but which requires less training time would be useful.
> The added paragraph is:
> " Note that although powerful, IPM-based GANs tend to more computationally demanding than other GANs. Certain IPM-based GANs use a gradient penalty (e.g. WGAN-GP, Sobolev GAN) which is very computationally costly and most IPM-based GANs need more than one discriminator update per generator update (WGAN-GP requires at least 5 \citep{WGAN-GP}). Assuming equal training time for D and G, every additional discriminator update increase training time by a significant 50\%.”
>
> We do provide more comparison images in the linked GitHub. However, the link (the footnote on p18) is hidden to retain anonymity for the review process. We transferred the GitHub to an anonymous version for the reviewers. Here are the full minibatch for the models generating 256x256 cats:
> https://github.com/anonymousconference/RGAN/tree/master/images/full_minibatch.
>
> We would like to note that we ran additional stability analyses for CIFAR-10 in the appendix. We will consider doing using more benchmarks next time. We are very limited in our computing capability, thus we decided to only use CIFAR-10 and CAT. Next time, we will consider using CAT and CelebA instead.

---

### Public Comment · (anonymous) · 2018-11-12
**Does (average) relativistic really matters?**

I was attracted by your works since you put your paper on Arxiv (and codes on github).
One primary concern: although you presented quite a lot experiments around relativistic loss functions, it seems hard to prove that relativistic helps generally.

As shown in Tab. 1, with the 1st hyper-parameters (which seems less stable than the 2nd ones), RSGAN+GP>LSGAN>RaLSGAN>RaSGAN>RSGAN>RaHingeGAN>SGAN>HingeGAN>WGAN+GP>RaSGAN+GP, it only demonstrated that R/Ra sometimes work well but sometimes don't, and when to apply average to loss function is really a mystery.
In a set of more stable hyper-parameters, you get a totally different order (where WGAN+GP is the best one).

It seems R/Ra is very sensitive to hyper-parameters, hence in my reproducibility training of RGAN/RaGAN is very unstable and the results are worse than standard GAN(s).

---

> ### Author Response · Authors · 2018-11-12
> **Yes, "(average) relativistic really matters"**
>
> This comment appears to be written in bad faith to influence negatively the reviewers. Even your title is a fake question suggesting that relativistic GANs are not useful. If it wasn't your intention, then this shows a lack of judgment, as you could have sent me (the first author) an email as everyone does. I will answer you, but only once.
>
> First, the results as you even show, point out that relativistic average variants are almost always better than their non-relativistic counterparts.
>
> Both sets of hyper-parameters are stable, set 1 is DCGAN hyper-parameters and it what most people use. What differentiate the second set of hyper-parameters is that it use 5 Discriminator update per generator update (n_d = 5). These settings are needed to make WGAN-GP perform properly. However, in practice, very few people use n_d = 5 because it would take forever to train. Considering researchers and AI engineers want to apply GANs to real-world hard problems in high dimensions, they cannot afford to wait 3 times longer (instead of 1 D update and 1 G update, we have 5 D updates and 1 G update; 3 times more) for the model to finish training and not even necessarily reach better results. This is why Self-attention GANs and BigGANs use Hinge loss with n_d = 1 or 2.
>
> You fail to mention that our approach reached better results than WGAN-GP while using only n_d = 1 (thus 3 times faster).
>
> The only scenario where we could not show better results when using relativistic GANs where in the challenging experiments with extremely unstable hyper-parameters (that no one uses in practice; see Appendix) in which Relativistic GANs didn't seem to perform better or worse on average.
>
> However, in very realistic and meaningful scenarios where one has high-resolution images and a small sample size (as companies generally do), Relativistic GANs perform amazingly well when non-relativistic GANs cannot even train past generating pure noise. Which is why we were told by many engineers and practitioners that without relativistic GANs, they could have been able to achieve their goals. See for example ESRGAN (https://github.com/xinntao/ESRGAN) which won a competition because of the use of Relativistic GANs.
>
> This shows that yes, "(average) relativistic really matter".

---

### Public Comment · ~Wenyi_Tang1 · 2019-01-04
**Reproducibility Report of ICLR Reproducibility Challenge**

We participate the ICLR Reproducibility Challenge and reproduce the experiments (part) of RGAN.

Relativistic discriminator (RGAN) is proposed to improve training stability and visual quality of wide variety of GANs. We conduct a series of experiments, aiming to reproduce the results of the RGAN. We compare frechet inception distance (FID) and inception score (IS) among 10 different loss functions (NSGAN, LSGAN, NSGAN-GP, WGAN-GP and the relativistic version) via 2 GAN architectures (DCGAN and ResNet) and 3 different normalization methods (batch normalization, spectral normalization and no normalization). Our tasks include generating 32x32 natural images (trained by CIFAR10) and 64x64 human faces (trained by CelebA).

We compare relativistic GAN and relativistic average GAN to their counter-parts in totally 20 cases. We found in most cases, relativistic GAN is an effective method to stabilize training process and can also improve image quality, while relativistic average GAN is not very effective and sometimes corrupt training in our experiments. Apply relativistic GAN with spectral normalization on discriminator is strongly recommended. Apply gradient penalty to discriminator loss function can further improves the image quality at the cost of higher computation.

In general, the experimental results show that RGAN is able to stabilize training process and improve generated image quality in 15 out of 20 cases, and fail to generate reasonable images (unconverged) in 2 cases. While the relativistic average GAN (RaGAN) can improve the quality in 9 out of 20 cases and fail to generate reasonable images in 4 cases. Our code is available at https://github.com/LoSealL/VideoSuperResolution.

---

> ### Author Response · Authors · 2019-01-04
> **interesting results!**
>
> It's very interesting as I have generally observed that RaGANs were better in almost every scenarios I tested. This is why replications are so useful :) .
>
> I'd be curious to see in which scenarios each approach worked best. The link is to VideoSuperResolution, is that the correct link?

---

> > ### Public Comment · ~Wenyi_Tang1 · 2019-01-06
> > **the link is correct**
> >
> > Yes, we use our VSR repo as a base-repo to develop RGAN models. You can find GAN architecture defined in VSR/Models/Gan.py. You can run the training and validation referring to README_ICLR.md.
> > Also, our PR to the challenge is at https://github.com/reproducibility-challenge/iclr_2019/pull/135 , where you can find the full report :)
> >
> > P.S: You may need to delete the dot "." when click on that link directly :)

---

### Meta-Review · Area_Chair1 · 2018-12-13
**A useful improvement for GAN training**

**Confidence:** 4
**Recommendation:** Accept (Poster)

**Metareview:**

All authors agree that the relativistic discriminator is an interesting idea, and a useful proposal to improve the stability and sample quality of GANs. In earlier drafts there were some clarity issues and missing details, but those have been fixed to the satisfaction of the reviewers. Both R1 and R3 expressed a desire for a more theoretical justification of why the relativistic discriminator should work better, but the empirical results are strong enough that this can be left for future work.